# OFFLINE VS. ONLINE LEARNING IN MODEL-BASED RL: LESSONS FOR DATA COLLECTION STRATEGIES

## ABSTRACT

Data collection is crucial for learning robust world models in model-based reinforcement learning. The most prevalent strategies are to actively collect trajectories by interacting with the environment during online training or training on offline datasets. At first glance, the nature of learning task-agnostic environment dynamics makes world models a good candidate for effective offline training. However, the effects of online vs. offline data on world models and thus on the resulting task performance have not been thoroughly studied in the literature. In this work, we investigate both paradigms in model-based settings, conducting experiments on 31 different environments. First, we showcase that online agents outperform their offline counterparts. We identify a key challenge behind performance degradation of offline agents: encountering Out-of-Distribution states at test time. This issue arises because, without the self-correction mechanism in online agents, offline datasets with limited state space coverage induce a mismatch between the agent's imagination and real rollouts, compromising policy training. We demonstrate that this issue can be mitigated by allowing for additional online interactions in a fixed or adaptive schedule, restoring the performance of online training with limited interaction data. We also showcase that incorporating exploration data helps mitigate the performance degradation of offline agents. Based on our insights, we recommend adding exploration data when collecting large datasets, as current efforts predominantly focus on expert data alone.

## 1 INTRODUCTION

Model-based Reinforcement Learning (MBRL) has emerged as a powerful paradigm, achieving state-of-the-art performance in complex tasks (Hansen et al., 2024; Hafner et al., 2023) and surpassing model-free methods (Schulman et al., 2017; Haarnoja et al., 2018) in both performance and sample efficiency. At the core of MBRL lies training a world model (Ha & Schmidhuber, 2018; Moerland et al., 2023; Morales, 2020) that captures the environment dynamics, which is task-agnostic and can generalize beyond learning mere action responses (Bruce et al., 2024). This model can then be leveraged for sampling-based planning (Hansen et al., 2024; Chua et al., 2018; Zhu et al., 2023) or training policies in imagination, eliminating the need for direct agent-environment interactions (Hafner et al., 2020; 2023).

The success of downstream tasks in MBRL critically depends on the accuracy of the world model (Asadi et al., 2019; Yao et al., 2021; Wang et al., 2024; Kidambi et al., 2020), which in turn relies heavily on the quality and diversity of training data (Mediratta et al., 2024; Suau et al., 2023; Kumar et al., 2022). This dependency raises the question: *What is the optimal strategy for data collection to effectively train these models?*

One popular strategy in robotics is to collect a large offline dataset of expert demonstrations to enable imitation learning, which essentially aims to match the distribution of expert actions (Fu et al., 2024; Collaboration & et al., 2023). However, collecting expert trajectories is expensive, less scalable, and not feasible for all tasks. As an alternative, training data can be collected online; this in turn requires an agent to interact directly with its environment during training. The actions the agent takes to collect the data can be driven by maximizing a task-reward (Hafner et al., 2023) or an exploration strategy by minimizing model uncertainty (Pathak et al., 2017; Sekar et al., 2020). This approach, while potentially more adaptive, incurs the cost of generating new data during training.

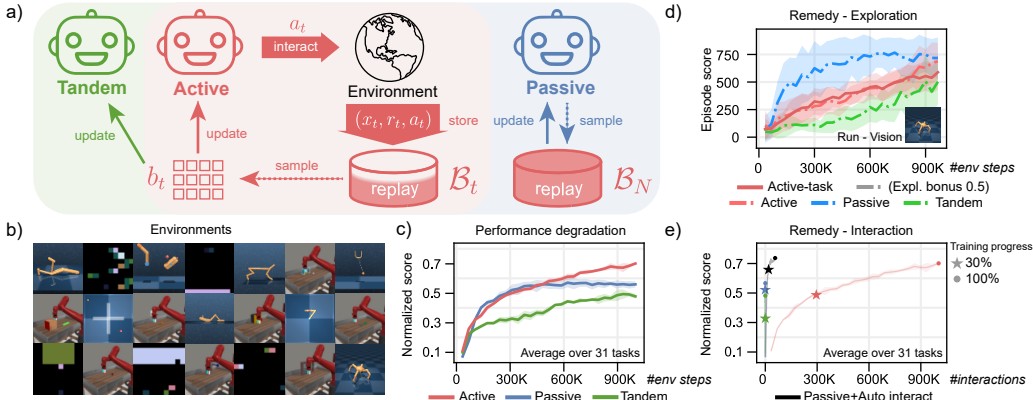

Figure 1: **Investigation of the performance degradation in offline agents and potential remedies.** a) Illustration of Active, Passive, and Tandem agents. The Active agent is trained using online RL and is allowed to interact with the environment. The Passive agent is trained from the full buffer of an Active agent, without performing any additional interactions. The Tandem agent, is also trained offline, but samples batches from the Active agent's replay buffer in the exact same sequence. b) We conduct experiments in 31 tasks across various domains. c) Illustration of the performance degradation in Passive and Tandem agents w.r.t. the Active agent. d-e) exploration data (d) and online interaction (e) effectively mitigate performance degradation observed in offline Passive agents.

While these two data collection paradigms (offline and online) are exhaustively studied in isolation, world models offer the unique advantage of integrating data from both paradigms, as the world model itself learns task-agnostic environment dynamics. Our work aims to provide a unified perspective on training world models from offline and online data in an MBRL setting by addressing two key questions: (1) How can we **leverage offline data** to train a robust world model and (2) **what combination of data collection strategies** yields the best performance at the lowest cost across different scenarios?

Previous works investigated these questions in the context of Q-learning and concluded that training an agent fully from offline data leads to degraded performance due to Out-Of-Distribution (OOD) queries of Q-functions (Ostrovski et al., 2021; Yue et al., 2023; 2022). While the degradation is also widely observed in offline MBRL, the coupling of the world model and policy presents a unique challenge in interpreting the degradation process. Current studies focus on proposing solutions on the premise of OOD-induced performance degradation (Yu et al., 2020; Kidambi et al., 2020; Wang et al., 2024) but lack a deep understanding of the failure process behind. Therefore, we investigate potential explanations of the degradation process and explore the effectiveness of common data-oriented strategies (Ostrovski et al., 2021; Yarats et al., 2022) in various tasks and domains from a unified perspective, which can provide valuable insights for future dataset collection.

To gain these insights, we employ DreamerV3 (Hafner et al., 2023) across diverse environments including locomotion, manipulation, and numerous other robotic tasks. As shown in Fig. 1, we examine three scenarios: (1) an Active agent training tabula rasa, (2) a Tandem agent replaying the learning history of the Active agent in the same temporal order but with a different random initialization, and (3) a Passive agent with access to the Active agent's full experience from the start, also with a different random initialization.

Our key findings reveal that in a task-oriented setting, Tandem and Passive agents underperform compared to the Active agent, primarily due to visiting novel states during evaluation. This OOD tendency stems from the absence of self-correction mechanism in offline agents, causing a mismatch between the agent's imagination and real rollouts, which misguides policy training. We demonstrate that using offline exploration data instead of solely task-oriented data mitigates this problem and, surprisingly, find that expert demonstrations alone are insufficient for high performance in MBRL. However, we showcase that performance can be recovered with minimal environment interactions. Based on these results, we analyze an adaptive fine-tuning agent that can recover the Active agent's performance with just $6\%$ of environment interactions relative to its offline dataset. As a result of our large-scale experimental study, we suggest to everyone collecting expert demonstration data to also collect exploration data for sufficient state-space coverage.

Our contributions are as follows:

- **Analysing the process behind performance degradation** in offline model-based agents, along with several practical considerations.
- **Demonstrating the benefits of exploration data** and proposing that a mixed reward function enhances state-space coverage in data collection, preventing performance degradation in offline training while maintaining strong task performance.
- **Examining world-model loss as a metric for targeted active data collection**, thereby substantially enhancing the efficiency of offline agents with minimal additional interactions.

## 2 METHOD

### 2.1 PRELIMINARIES

**Model-based Reinforcement Learning** In this work, we consider environments that can be described by a partially observable Markov Decision Process (POMDP), with high-dimensional observations $x_t$, which are encoded into latent representations $s_t$, state-conditioned actions $a_t$ generated by an agent and scalar rewards $r_t$ (conditional on $s_t$ and $a_t$) generated by the environment. In MBRL, our aim is to learn the latent transition dynamics by a **world model** $\hat{\mathcal{T}}(s_{t+1} \mid s_t, a_t)$ and find an optimal **policy** $\pi(a_t|s_t)$ maximizing the expected discounted return with discount factor $\gamma$:

$$\pi^* = \arg\max_{\pi} \mathbb{E}_{\substack{s_t \sim \hat{\mathcal{T}}(\cdot|s_{t-1},a_{t-1}) \\ a_t \sim \pi(a|s_t)}} \left[ \sum_{t=0}^{\infty} \gamma^t r(s_t, a_t) \right]. \tag{1}$$

**DreamerV3** We use DreamerV3 (Hafner et al., 2023), a state-of-the-art model-based RL method, as the base architecture in all our experiments. Based on the Recurrent State-Space Model (RSSM) (Hafner et al., 2018) summarized in Eq. (2), the world model predicts the latent state $s_t = (h_t, z_t)$ from the previous state and action, where $h_t$ is the deterministic and $z_t$ is the stochastic state component. The estimated observation $\hat{x}_t$, reward $\hat{r}_t$, and continuation flag $\hat{c}_t$ (signalling whether the episode has ended or not) are decoded from the latent states; given by the tuple $\hat{e}_t = (\hat{x}_t, \hat{r}_t, \hat{c}_t)$. The policy has an actor-critic architecture, detailed in Eq. (3). $R_t$ is the discounted return from state $s_t$. For the off-policy updates of DreamerV3, environment interactions are added to a replay buffer $\mathcal{B} = \{(x_t, a_t, r_t, c_t, \dots)\}_{t=1}^N$, where each tuple contains the observation $x_t$, action $a_t$, reward $r_t$, continuation flag $c_t$, and optionally other variables collected from the environment.

$$\begin{array}{llll} \textbf{Sequence model:} & h_t = f_\phi(h_{t-1}, z_{t-1}, a_{t-1}) & \textbf{Encoder:} & z_t \sim q_\phi(z_t \mid h_t, x_t) \\ \textbf{Dynamics predictor:} & \hat{z}_t \sim p_\phi(\hat{z}_t \mid h_t) & \textbf{Decoder:} & \hat{e}_t \sim p_\phi(\hat{e}_t \mid h_t, z_t) \end{array} \tag{2}$$

$$\textbf{Actor:} \quad a_t \sim \pi_\theta(a_t \mid s_t) \qquad \textbf{Critic:} \quad v_\psi(s_t) \approx \mathbb{E}_{p_\phi, \pi_\theta}[R_t] \tag{3}$$

DreamerV3 minimizes the world model loss, which is a weighted loss of multiple components and is defined in the original paper (Hafner et al., 2023), as shown in Eq. (4).

$$\mathcal{L}(\phi) \doteq \mathbb{E}_{q_\phi} \left[ \sum_{t=1}^T (\beta_{\text{dyn}} \mathcal{L}_{\text{dyn}}(\phi) + \beta_{\text{rep}} \mathcal{L}_{\text{rep}}(\phi) + \beta_{\text{pred}} \mathcal{L}_{\text{pred}}(\phi)) \right]. \tag{4}$$

It consists of the dynamics-based loss components given by $\mathcal{L}_{\text{dyn}}$ and $\mathcal{L}_{\text{rep}}$, defined in Eq. (S1), as well as the loss $\mathcal{L}_{\text{pred}}$ from three prediction heads: observation reconstruction, reward estimation, and continuity prediction.

The following three-step cycle is repeated throughout the training process of DreamerV3: (1) The agent interacts with the environment to collect data, adding it to its replay buffer $\mathcal{B}$. Meanwhile, the latent states $(h_t, z_t)$ are updated closed-loop using the current observation $x_t$ and are used to compute the action. (2) The world model is trained on a batch of sequence data uniformly sampled from the replay buffer using the loss function shown in Eq. (4). (3) Open-loop trajectories are generated in imagination by the world model to train the actor and critic networks.

### 2.2 LEARNING AGENTS

In order to investigate the online and offline training paradigms, we design three off-policy agents, as shown in Fig. 1, each representing a different variation of training data collection.

**Active agent** is the typical RL agent in online RL. It interacts with the environment and performs training steps using the collected data by its own policy. An Active agent can adapt its world model with its own policy rollouts, which is a self-correction mechanism, enabling the agent to learn from its own mistakes (Ostrovski et al., 2021).

**Passive agent** is trained offline without any environment interactions by uniformly sampling data from the *final* replay buffer $\mathcal{B}_N$ of an Active agent. This gives the Passive agent access to the full data of the Active agent right from the start of the training process, including high-reward trajectories.

**Tandem agent** is another agent trained offline, but sees the training data in the same order as the Active agent, i.e. the training batches $b_t$ are replayed exactly as they were sampled during the training of the Active agent (Ostrovski et al., 2021). The goal here is to introduce a more controlled offline learning setting than the Passive agent, with the only difference from the Active agent being the model initialization. This setup facilitates easier interpretation of the experimental results.

The offline agents, Passive and Tandem, are initialized independently of the Active agent used for data collection with a different random seed. The pseudocode of the agents is in Appendix A.1.5.

## 3 EXPERIMENTS

We use DreamerV3 for all our experiments (details on hyperparameters can be found in Appendix A.1). In total, we conducted 2000 experiments using 20 000 GPU hours. All agents are trained from scratch using task-oriented rewards unless specified otherwise.

### 3.1 ENVIRONMENT SETUP

Our experiments are conducted in the Deepmind Control Suite (DMC) (Tunyasuvunakool et al., 2020; Yarats et al., 2022), Metaworld (Yu et al., 2019), and MinAtar (Young & Tian, 2019) domains, including a total of 31 tasks. These are representative environments for robotic locomotion, manipulation, and discrete game tasks. The environment settings mainly follow the default settings in Hafner et al. (2023). The results for all individual experiments and detailed setups are provided in the Appendix A.8 and Appendix A.1. Whether state or image observations are used is indicated alongside the task name as "proprio" or "vision" respectively. We run 1 million environment steps per task, training every second step, with results averaged across three seeds unless stated otherwise. For the Passive and Tandem agents, we keep the same total number of environment and training steps as the Active agent to ensure consistency and comparability; however, without collecting any interaction data, as explained in Appendix A.1.4.

### 3.2 METRICS FOR ANALYSIS

**World model loss** The mean error of the world model for the prediction of dynamics, observation, reward, and continuity (Sec. 2.1). It is an indicator of the total aleatoric and epistemic model uncertainty and can serve as a simple OOD measure (Yu et al., 2020; Chen et al., 2023).

**Episode score** The undiscounted sum of rewards over the episode.

The metrics shown in all figures are calculated as follows, unless specified otherwise: (1) Every 5K environment steps, we roll out the agent's policy for a total of 4 episodes. (2) We compute the mean episode score and the mean world model loss across the 4 episodes. Each agent is evaluated in an on-policy manner on its own test-time trajectories. The data distributions of visited states are thus conditioned on the policy and are different for individual agents.

### 3.3 TOY EXAMPLE

We first study the performance of all learning agents in a toy environment. We select the point mass maze environment in DMC, where an actuated 2-DoF point mass has to reach the red goal position, as shown in Fig. 2. The results show that only the Active agent successfully solves the task, while both agents trained offline fail, showing degraded performance compared to the Active agent.

**Hypothesis: Lack of self-correction causes OOD errors** The policy in DreamerV3 is trained purely in the imagination of the world model. As a result, the policy can learn to exploit inaccuracies

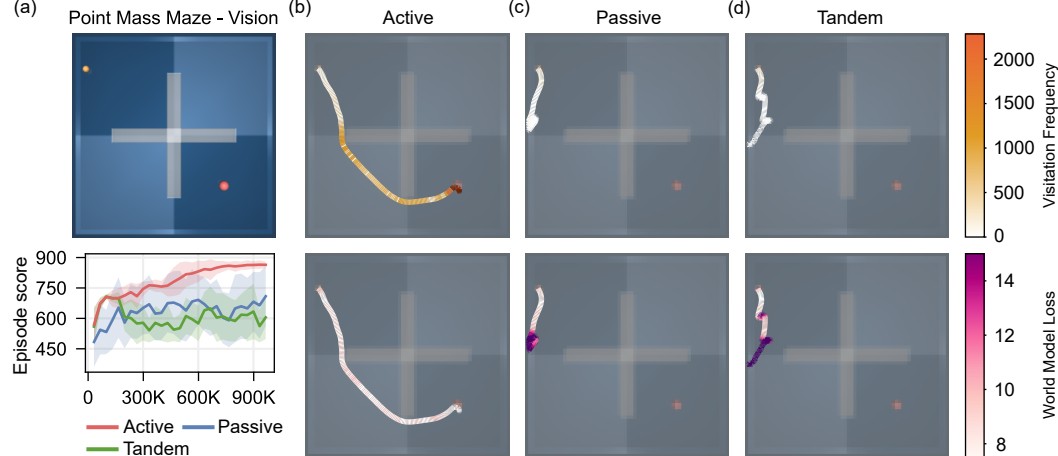

Figure 2: **Example of the degraded performance during offline training in 2D point mass maze environment.** The task is to move the yellow point mass from the top-left initial position to the red marker in the bottom-right of the maze, which is the goal position. The episode score of each agent is shown in (a). In (b-d), we show the point mass trajectory generated by the final model after 1M environment steps. The two heatmaps on the trajectory represent: 1) a count-based frequency of each covered cell that is visited in the replay buffer and 2) world model loss on each visited state. The median visitation frequency along the shown trajectory is 608.5 for Active, 12.5 for Passive, and 9.0 for Tandem.

in the imagination. The Active agent continuously collects data from regions where the world model could be unreliable, specifically for regions where the world model predicts a high reward and, therefore, the policy is likely to visit. Training the world model on the collected data from these regions helps to improve the world model in a targeted manner with respect to the current Active agent's policy. This not only helps to improve the policy to solve the task but also makes the world model adapt to the agent's policy rollouts, ensuring sufficient data coverage around its self-rollouts. Consequently, the agent is unlikely to encounter novel states when rolling out the policy during evaluation.

The agents trained offline lack this critical feedback loop of self-correction. Although the overall training data distribution is the same as the Active agent, differences in sampling sequences (Passive) and/or model initializations (Passive and Tandem) lead to distinct policies during training. To effectively improve these policies, the training data generated from the world model's imagination should closely match real rollout performance. However, without self-correction and constrained by data coverage tailored to another agent's policy, the imagination of this limited-capability world model fails to align with real rollouts under its own policy, leading to a persistent discrepancy between imagination and reality in offline training. Consequently, the policy will exploit these inaccuracies during training and be updated blindly to eventually steer the agent toward novel, unvisited areas. During test time, visiting novel states can lead to world model prediction errors and, therefore, suboptimal policy actions. It creates a catastrophic cycle where each compromised action leads to further novel states and additional inaccuracies in the world model until the episode ends or the agent accidentally re-enters into a familiar state.

We observe this behavior in the performance of the three agents as shown in Fig. 2. The Active agent learned to adapt its world model to its own rollouts; therefore, it did not meet any novel states when rolling out the policy for evaluation, as shown by the consistent low world model loss and high visitation frequencies alongside its trajectory. However, this is not the case for the Passive and Tandem agents. From the start, their policies seem to behave anomalously, guiding them towards a suboptimal direction even in the regions familiar to the world model. Since the task-oriented dataset has limited state-space coverage, they inevitably visit novel states, and their mistakes are catastrophic. As a result, both the Passive and Tandem agents cannot recover and end up in OOD states until the end of the episode, failing to solve the task.

To summarize, **self-correction ensures sufficient data coverage related to the agent's policy rollouts**, thereby **1) preventing OOD errors** and **2) facilitating policy training** by reducing gaps between imaginations and real rollouts. Without self-correction, imagination gaps compromise policy

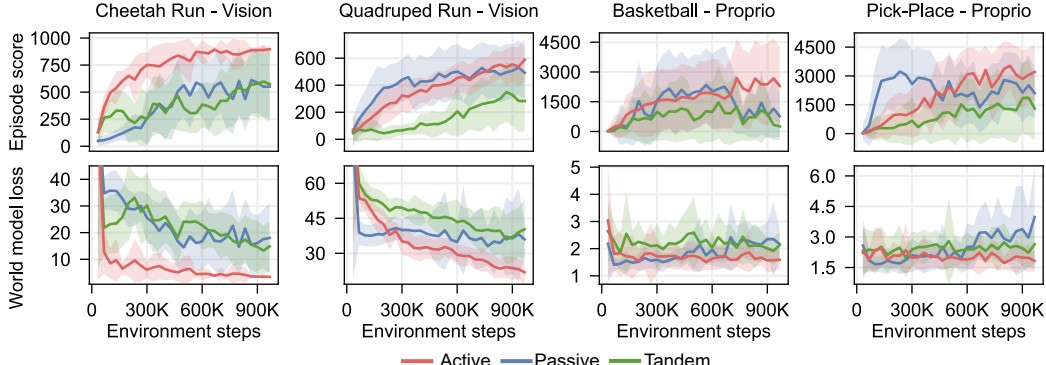

Figure 3: **Episode score and world model loss during evaluation rollouts of 4 selected tasks.** The first two are from DMC and the last two are from the Metaworld domain. The performance degradation of offline agents, including Passive and Tandem, is common across domains and tasks, especially for Tandem agents.

training and push offline agents toward OOD states, where they become trapped in a catastrophic cycle that leads to further performance degradation.

Our hypothesis is generally in line with previous research in model-free RL (Ostrovski et al., 2021; Yue et al., 2023; Emedom-Nnamdi et al., 2023; Kumar et al., 2020b) , which attributes performance degradation to extrapolation errors in Q-values in OOD state-action pairs during training and evaluation. However, in the context of MBRL, the paradigm is shifted from a focus on Q-functions to the coupling of a world model and a policy network.

### 3.4 VALIDATION ACROSS TASKS

The performance degradation phenomenon in offline agents is observed across various tasks and domains, as shown in Fig. 3 and Appendix A.8.2. In tasks such as *Quadruped Run - Vision* and *Pick-Place - Proprio*, the Passive agent initially demonstrates a faster increase in performance but has a larger variance or even experiences performance drops as training progresses. The degraded performance in Passive and Tandem agents is accompanied by a significantly larger world model loss on evaluation episodes than the Active agent. Given that a high world model loss indicates novel states, this observation supports our hypothesis in Sec. 3.3. The discrepancy between imagined and real rollouts in offline agents is shown in Appendix A.4. Our detailed inspections on a timestep level in Appendix A.5.1 further validate our hypothesis of the catastrophic cycle during testing. Fig. 3 also shows a potential advantage of Passive agents: faster convergence by having access to high reward trajectories from the start of training (validated in Appendix A.7), though additional measures may be necessary to ensure training stability. The results of Tandem agents also follow the findings of degraded performance of the Tandem training regime in Ostrovski et al. (2021) and extend its validity to MBRL.

### 3.5 DEEP DIVE INTO PERFORMANCE DEGRADATION

#### 3.5.1 OOD IN MBRL

**Both world model and policy affect performance degradation** To investigate which one, world model or policy, plays the most important role in causing the performance degradation, we carry out a more controlled experiment in Fig. S5, with the detailed description in Appendix A.6. By using an identical world model in Passive or Tandem agents to their Active counterparts, we disentangle the compounding effect from the world model and policy. The results show that deviations in both the world model and policy from the Active agent contribute to performance degradation, with their relative impacts depending on the specific task.

**What is the difference to supervised learning?** In classical supervised learning, a model is optimized on an offline dataset, e.g., for image classification. Training on independent and identically distributed data from different random initializations typically yields similar performance, showing robustness to initialization. Why is this not the case in the MBRL setting, where Tandem agents perform worse than Active agents, despite one expecting the world model to perform equally well across seeds given the same data? This is because offline trained agents will cause states to be visited during policy optimization that are not collected by the Active agent, leading to OOD queries to the model.

### 3.5.2 WORLD MODEL LOSS IS A PESSIMISTIC INDICATOR OF PERFORMANCE DEGRADATION

The world model loss is due to prediction errors arising from both epistemic and aleatoric uncertainty. Novel states lead to high variance predictions due to epistemic uncertainty induced by insufficient state space coverage during training. Overlaid are errors due to partial observability and environment stochasticity. In particular, the latter factors can lead to high model loss without significant impacts on performance, depending on whether exact predictions are required for the task at hand. In addition, even when the agent is in novel states, other factors, e.g. environment constraints, and the policy producing correct actions by coincidence in hallucinations of the world model, can reduce the impact of a poorly performing world model on agent performance. Therefore, the world model loss is a pessimistic indicator of performance degradation.

### 3.5.3 EXPERT DATA ALONE EXACERBATES OOD ISSUES

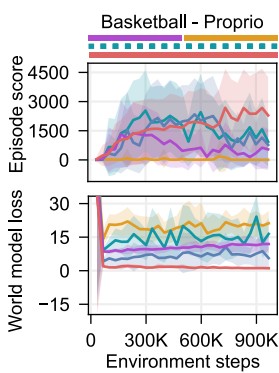

Expert data is commonly used in offline learning, but compared to data collected by the Active agent, its coverage is more limited to task-specific trajectories, typically capturing only certain ways of solving the task. As a result, states are more likely to be OOD for the world model, resulting in even worse task performance, as shown in Fig. 4, where we treat the second half of the buffer as expert data. As expected, the world model loss evaluated on test-time trajectories is significantly larger than for other agents with suboptimal or mixed data. For more details, see Appendix A.7.

### 3.5.4 CONSIDERATIONS IN PRACTICAL APPLICATIONS

In further experiments, we find that initializing the Passive agents' weights identically to the Active agents' does not improve task performance. In contrast, even minor differences in the model initialization of Tandem agents compared to Active agents leads to degraded performance, reflecting the chaotic training dynamics of gradient-based optimization. See Appendix A.7 for more details.

Figure 4: **Performance comparison of Active, Passive as well as Passive agents trained on expert, suboptimal, and mixed data**, which is implemented by splitting the replay buffer of the Active agent in different ways.

## 4 POTENTIAL REMEDIES FROM A DATA PERSPECTIVE

Based on the previous analysis, we conclude that insufficient state coverage during training of Passive and Tandem agents leads to worse model performance, which results in visiting OOD states during evaluation. To address this, we propose two strategies for effective agent training with offline datasets: **training on an exploration dataset** and **(adaptively) incorporating self-generated data**.

### 4.1 TRAINING ON EXPLORATION DATA

We investigate how training on exploration data affects the performance of Active, Passive and Tandem agents. Here, we use Plan2Explore (Sekar et al., 2020), where the objective is to maximize the

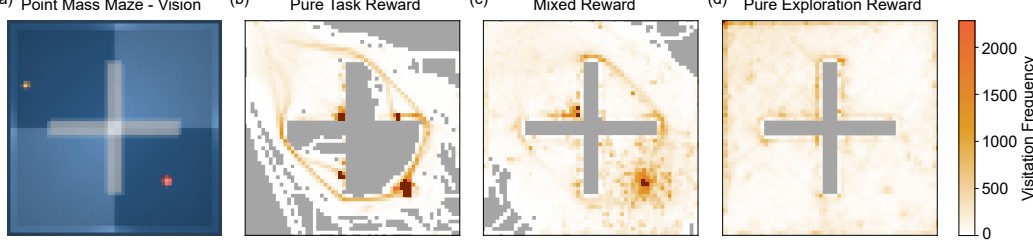

Figure 5: **State visitation in the Point Mass Maze task.** They are calculated using the discretized states from three different Active agents' final replay buffers after 1M environment steps. (b) Agent in a pure task-oriented setting. (c) Agent with pure exploration rewards based on ensemble disagreement (Sekar et al., 2020). (d) Agent with a mixed reward: task plus exploration rewards, see Eq. (5) with $w_{\text{expl}} = 0.5$. The unvisited areas are painted gray, and the outliers that have extremely high values are painted dark red. Here the task-oriented agent only explores limited state space in the map and always follows certain routes towards the goal position, while the two explorative agents visit all the regions much more equally.

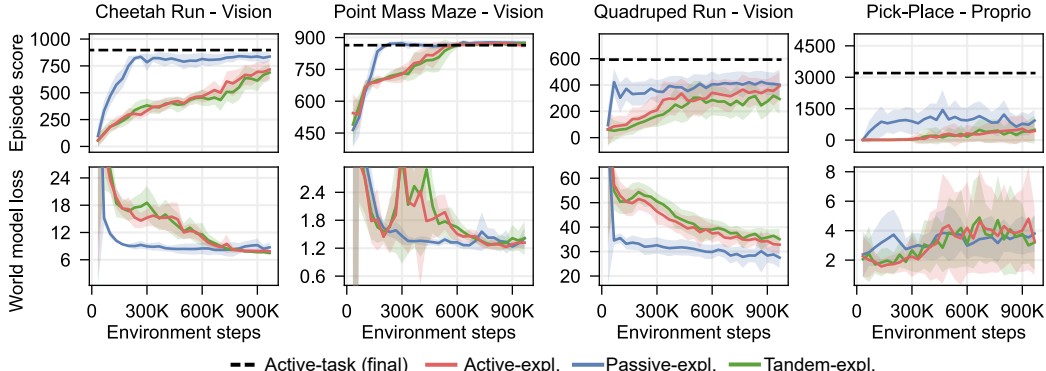

Figure 6: **Performance comparison when training on pure exploration data.** The dataset is generated by the Active-expl. agent with a behavioral policy based on ensemble disagreement (Sekar et al., 2020). We additionally show the baseline performance of a task-oriented Active agent.

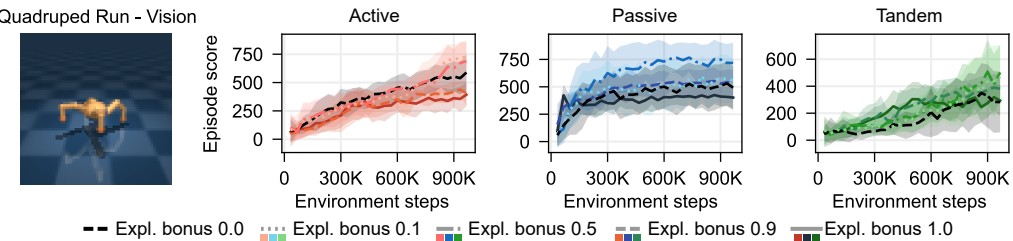

Figure 7: **Training on pure exploration data is not optimal.** Performance comparison when assigning different exploration bonuses $w_{\text{expl}}$ in the reward function. The black dashed lines represent pure task-oriented policy without any exploration bonus.

information gain of the world model. The exploration reward is calculated as ensemble disagreement, denoted by $r_{\text{disag}}$. We investigate exploration in two modes: 1) pure exploration in a task-free setting, i.e. agent only maximizes for $r_{\text{disag}}$, 2) a mixed reward setting, where $r_{\text{disag}}$ is added as an exploration bonus on top of the task reward:

$$r_t \doteq w_{\text{task}} \cdot r_{\text{task}} + w_{\text{expl}} \cdot r_{\text{disag}}, \tag{5}$$

where $w_{\text{task}}$ and $w_{\text{expl}}$ weights are normalized such that they sum up to 1.

For agents trained offline, exploration data in the training set can provide a larger state-space coverage, which can counteract the missing self-correction mechanisms of an active agent. Fig. 5 demonstrates how task-oriented data is narrower compared to exploration data. The addition of exploration data becomes crucial in alleviating the OOD challenge during evaluation, as validated in Fig. 6, where the training data is gathered by an Active agent based on pure exploration rewards $r_{\text{disag}}$. As a result, the Passive agents consistently outperform the Active, and the performance of the Tandem agents matches their Active counterparts. Furthermore, the relationship between task performance and world model loss generally also matches the findings in Sec. 3.4. However, some cases in Appendix A.8.4 indicate that world model loss can occasionally be less predictive of task performance. This inconsistency arises as novel regions for the world model shrink with exploration data, leading to lower world model losses. In addition, the pure exploration dataset contains numerous trajectories irrelevant to the task, reducing the world model's accuracy in task-specific states and preventing the effective learning of the task policy. Consequently, task performance becomes increasingly dependent on the task difficulty. For example, in two challenging tasks – *Quadruped Run - Vision* and *Pick-Place - Proprio* – their overall performance is significantly lower than that of the task-oriented version, as shown in Fig. 6.

To this end, we investigate the mixed reward setting, where we add the exploration reward as a bonus, as defined in Eq. (5). This approach allows a more concentrated exploration near the goal, as shown in Fig. 5, preventing the excessive exploration of irrelevant areas that could arise from a purely explorative dataset.

Indeed, in Fig. 7, we show that pure exploration is hardly the best option for the hard tasks like *Quadruped Run - Vision*. The addition of an exploration bonus e.g. $w_{\text{expl}} = 0.5$ together with task rewards in *Quadruped Run - Vision* can lead to an improved task performance compared to runs with

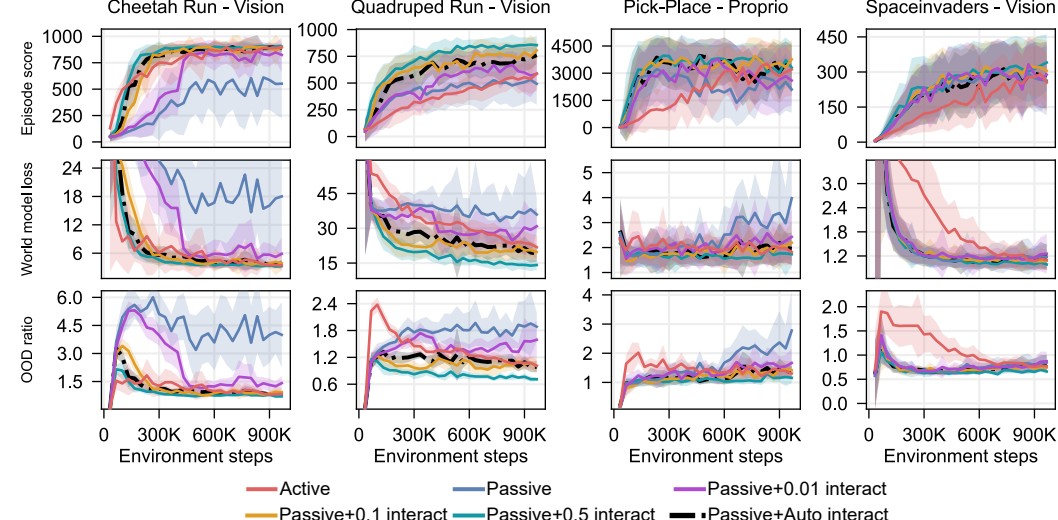

Figure 8: **Performance comparison when allowing adding additional self-generated data for Passive agents.** The Passive+Auto interact agent adds 6.5% self-generated data in Cheetah Run - Vision, 2.9% in Quadruped Run - Vision, 9.8% in Pick-Place - Proprio, and 0.5% in Spaceinvaders. The percentage is calculated w.r.t. to the size of the final replay buffer of Active agents.

pure task rewards, especially in Passive agents. A downside of this approach is the introduction of the hyperparameter $w_{\text{expl}}$, the optimal value of which can depend on the specific task as shown in our experiments in Appendix A.8.1.

## 4.2 ADDING ADDITIONAL SELF-GENERATED DATA

We have demonstrated the critical importance of self-correction. However, as training solely on interaction data is expensive, and offline data is often cheaply available; we would like to explore how one can most effectively combine fixed offline data with online interaction data. To analyze this interplay, we first examine a strategy that uses a predetermined schedule for the Passive agent to interact with its environment.

Specifically, for every $N$ environment steps, the Passive agent is allowed to collect 2K-step transitions based on its learned policy. Then the interactive data will be added to expand the replay buffer for later sampling during world model training as usual. By choosing a different $N$, we can adjust the frequency of interactive data injection. Experiments were conducted with $N$ set to 4K, 20K, and 200K, respectively corresponding to 50%, 10%, and 1% self-generated data. The results are shown in Fig. 8. Accordingly, merely 10% additional self-generated data can already result in a significant improvement in the episode score as well as a notable reduction in the world model loss, recovering the performance of its Active counterpart. In certain environments, such as the *Spaceinvaders* from the MinAtar domain, the Passive agents may already solve the task and have a faster convergence than the Active one; therefore, self-generated data provides no performance increase.

**Adaptive interaction** Upon examining the results with a fixed schedule, we see that interaction ratios to restore agents' performance vary across tasks. Therefore, we analyze an adaptive interaction schedule based on the insights of OOD states causing degenerate performance. We calculate a ratio by dividing the world model loss on evaluation trajectories by the loss on trajectories in the replay buffer. This ratio measures the novelty of the trajectories visited by the current learned policy compared to those seen during training and enables a single threshold for adding self-generated data across tasks.

We set the threshold for the OOD ratio to 1.35 (see the ablation study in Appendix A.1.6) and inspect it every 5K environment steps over 4 evaluation episodes. If the OOD ratio exceeds this threshold, the Passive agent collects 2K-step transitions from the environment using its learned policy, denoted as Passive+Auto interact (refer to Appendix A.1.5 for the agent's pseudocode). As shown in Fig. 8, this strategy fine-tunes self-generated data injection based on task demands, achieving similar performance with less data (5.67% across 31 tasks) compared to an agent that regularly adds 10% self-generated data. The inspection frequency can be reduced to lower evaluation costs. For more results, see Appendix A.8.3. A complete offline evaluation would be desirable, but is outside the scope of this paper. We hope to inspire research in this direction.

## 5 RELATED WORK

**Performance Degradation in Offline Model-based Agents**     Performance degradation of offline agents is a known phenomenon in MBRL (He, 2023) and is mainly attributed to two factors:

**1) The distribution mismatch between training data and the states visited by the learned policy** (Kidambi et al., 2020; Chen et al., 2023; Yu et al., 2020; Cang et al., 2021). These inaccuracies in the world model within unseen regions are then exacerbated by compounding errors in multi-step predictions (Asadi et al., 2019; Janner et al., 2019). These accumulated errors in the model-based imagination process based on OOD queries can mislead both policy training (Wang et al., 2024) and planning by overestimation in critics (Sims et al., 2024), ultimately resulting in a performance drop.

**2) The inability of offline agents to self-correct through active data collection** (He, 2023; Cang et al., 2021; Yu et al., 2020). Prior works on offline agents (Ostrovski et al., 2021; Tang et al., 2024; Emedom-Nnamdi et al., 2023; Lin et al., 2024) have shown that utilizing data from interactions with the environment introduces a corrective feedback loop (Kumar et al., 2020a), allowing the agent to learn from its own mistakes and consequently improve its task performance.

Building on existing studies, we explore phenomena across various tasks and domains in model-based RL using DreamerV3. Additionally, we investigate the conditions (e.g. the nature and quality of the dataset) that exacerbate distribution mismatches and model inaccuracies.

**Remedies to Support Offline Training**     To address performance degradation in offline model-based agents, many studies add conservatism to their algorithms. One method is to include an uncertainty penalty in the reward function to deter the agent from exploring new states (Kidambi et al., 2020; Yu et al., 2020; 2021; Wang et al., 2024), while another employs trust-region updates to maintain the learned policy's proximity to the data collection policy (Matsushima et al., 2021). RAMBO (Rigter et al., 2022) trains an adversarial environment model that generates pessimistic transitions for OOD state-action pairs, reducing the value function in uncertain regions. In contrast, MAPLE (Chen et al., 2023) enables adaptive agent behavior in OOD regions during deployment, using a context-aware policy based on meta-learning techniques.

While these methods provide insights on mitigating performance degradation in offline MBRL, few address which type of data best facilitates offline training. In model-free RL, studies suggest adding self-generated data (Ostrovski et al., 2021; Lee et al., 2021) and emphasize the importance of diversity and exploration (Mediratta et al., 2024; Suau et al., 2023; Kanitscheider et al., 2021; Kim et al., 2023). We extend these ideas to model-based RL with validation in various tasks and domains.

## 6 CONCLUSIONS AND DISCUSSIONS

In this work, we show that visiting novel states during evaluation is the key factor behind the degradation of the performance of offline model-based agents through a wide range of experiments across various domains. From a data perspective, we identify that training on partially exploratory data collected using a mixed task-exploration reward function is effective in mitigating performance degradation. Importantly, training offline solely on expert data exacerbates performance degradation compared to a typical mixed dataset due to severe OOD issues. Additionally, our experiments show that adding as little as 10% self-generated data at regular intervals can significantly enhance the performance of Passive agents. When we allow the Passive agent to adaptively interact based on its world-model loss as a proxy measure of OOD state visitation, we observe a significant performance improvement while minimizing the need for additional interaction data. However, our method still requires evaluation rollouts. An offline measure would be desirable and is left for future research.

Overall, we highlight the importance of sufficient state-space coverage in the training data to train a robust model-based agent, which can be achieved either by an explorative offline dataset or by enabling the agent to learn from its own mistakes. As efforts to collect large-scale real-world data for robotics are increasing, the question arises: What is the best way to collect data to facilitate robust agent training? As model-based RL shows strong task performance and promises efficient fine-tuning and good transfer capabilities for new tasks, we suggest that dataset collection should incorporate exploration data. We plan to extend our experiments to other RL methods and real-world scenarios to identify optimal data collection strategies. We believe that our insights can help design a data-efficient fine-tuning method for robotics foundation models. This will help develop more resilient and adaptable agents capable of performing reliably in complex environments.

## 7 REPRODUCIBILITY STATEMENT

The hyperparameters used in our experiments are detailed in Appendix A.1. We control the randomness of each run—e.g., in environment initialization and model optimization—by setting fixed random seeds in our implementation. The code and generated datasets will be made publicly available upon acceptance of the paper. The results presented in our paper can be directly reproduced using the provided codebase without any additional modifications.

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

# A  APPENDIX

## A.1  IMPLEMENTATION DETAILS

### A.1.1  RUNTIME OVERVIEW

Our experiments comprised approximately 2000 runs, totaling 20000 GPU hours. Each run took between 8 and 15 hours, depending on the specific task. All experiments were conducted using NVIDIA RTX 4090 or A100 GPUs.

### A.1.2  MODEL HYPERPARAMETERS

For all experiments, we use the same model size $S$, defined in Hafner et al. (2023). Each agent, which consists of a world model, an actor network, and a critic network, has a total of 18M optimizable variables. We follow the default values in Hafner et al. (2023) for the training hyperparameters e.g. learning rate and batch size for each component of the agent as well as other hyperparameters. For more details about DreamerV3, please refer to Hafner et al. (2023).

### A.1.3  ENVIRONMENT HYPERPARAMETERS

We list the environment hyperparameters in Tab. S1. The implementation of the task *Point Mass Maze* is based on Yarats et al. (2022).

### A.1.4  ENVIRONMENT STEPS IN OFFLINE AGENTS

Tracking performance metrics relative to environment steps during online training is standard practice in the RL community. This methodology is also applied in the analysis of the offline Tandem agent in Ostrovski et al. (2021), which closely mirrors the behavior of its Active counterpart.

However, the Passive agent—by definition—does not interact with the environment and thus cannot influence environment steps. This poses a challenge for directly comparing its performance with that

Table S1: Environment hyperparameters for each domain

| Hyperparameter | DMC | Metaworld | MinAtar |
|---|---|---|---|
| Image Size | [64,64] | [64,64] | [32,32] |
| Action Repeat | 2 | 2 | 1 |
| Episode Truncate | - | - | 2500 |
| Parallel Env Num | 4 | 4 | 4 |

of the Active and Tandem agents. To ensure comparability across training procedures, we allow the Passive agent to interact with the environment during training in the same manner as an online agent, but without adding the resulting interaction data into its replay buffer. This setup enables the Passive agent to remain trained solely on an offline dataset while allowing performance comparisons based on environment steps, with only minimal code changes required.

### A.1.5 PSEUDOCODE OF METHODS

We add the pseudocode of the Active, Passive, and Tandem agents (in Alg. 1) as well as the second remedy (in Alg. 2) for better clarity.

---

**Algorithm 1** Learning agents (key difference is highlighted in its representative colors).

| **Active Agent** | **Passive Agent** | **Tandem Agent** |
|---|---|---|
| 1: **Initialize:** Replay buffer $\mathcal{B}$ = a few random episodes. | 1: **Initialize:** Replay buffer $\mathcal{B}$ = previous final $\mathcal{B}_A$. | 1: **Initialize:** Replay buffer $\mathcal{B}$ = previous final $\mathcal{B}_A$. |
| 2: World model $M$ + Policy $\pi$ by seed $S_A$. | 2: World model $M$ + Policy $\pi$ by seed $S_P$. | 2: World model $M$ + Policy $\pi$ by seed $S_T$. |
| 3: **for** each step $i$ **do** | 3: **for** each step $i$ **do** | 3: **for** each step $i$ **do** |
| 4:     Sample $\mathcal{D}_A^i \sim \mathcal{B}$ | 4:     Sample $\mathcal{D}_P^i \sim \mathcal{B}$ | 4:     Copy $\mathcal{D}_T^i = \mathcal{D}_A^i$ |
| 5:     Update $M$ using $\mathcal{D}_A^i$ | 5:     Update $M$ using $\mathcal{D}_P^i$ | 5:     Update $M$ using $\mathcal{D}_T^i$ |
| 6:     Train $\pi$ in the imagination of $M$ | 6:     Train $\pi$ in the imagination of $M$ | 6:     Train $\pi$ in the imagination of $M$ |
| 7:     Execute $\pi$ in the env to expand $\mathcal{B}$ | - | - |
| | - | - |
| 8: **Return:** Final $\mathcal{B}_A$, $\pi$ | 7: **Return:** Final $\mathcal{B}$, $\pi$ | 7: **Return:** Final $\mathcal{B}$, $\pi$ |

---

**Algorithm 2** Passive agents adding additional self-generated data (key difference is highlighted in its representative colors)

| **Passive Agent** | **Fixed Schedule** | **Adaptive Schedule** |
|---|---|---|
| 1: **Initialize:** Replay buffer $\mathcal{B}$ = previous final $\mathcal{B}_A$. | 1: **Initialize:** Replay buffer $\mathcal{B}$ = previous final $\mathcal{B}_A$. | 1: **Initialize:** Replay buffer $\mathcal{B}$ = previous final $\mathcal{B}_A$. |
| 2: World model $M$ + Policy $\pi$ by seed $S_P$. | 2: World model $M$ + Policy $\pi$ by seed $S_P$. | 2: World model $M$ + Policy $\pi$ by seed $S_P$. |
| 3: **for** each step $i$ **do** | 3: **for** each step $i$ **do** | 3: **for** each step $i$ **do** |
| 4:     Sample $\mathcal{D}^i \sim \mathcal{B}$ | 4:     Sample $\mathcal{D}^i \sim \mathcal{B}$ | 4:     Sample $\mathcal{D}^i \sim \mathcal{B}$ |
| 5:     Update $M$ using $\mathcal{D}^i$ | 5:     Update $M$ using $\mathcal{D}^i$ | 5:     Update $M$ using $\mathcal{D}^i$ |
| 6:     Train $\pi$ in the imagination of $M$ | 6:     Train $\pi$ in the imagination of $M$ | 6:     Train $\pi$ in the imagination of $M$ |
| - | 7:     **if** $i\%N == 0$ **then** // $N = 4K, 20K, 200K$ | 7:     **if** $i\%2K == 0$ and ood_ratio$_i$ > thres. **then** |
| - | 8:       Execute $\pi$ in the env to expand $\mathcal{B}$ by 2K step data | 8:       Execute $\pi$ in the env to expand $\mathcal{B}$ by 2K step data |
| - | | |
| - | | |
| 7: **Return:** Final $\mathcal{B}$, $\pi$ | 9: **Return:** Final $\mathcal{B}$, $\pi$ | 9: **Return:** Final $\mathcal{B}$, $\pi$ |

---

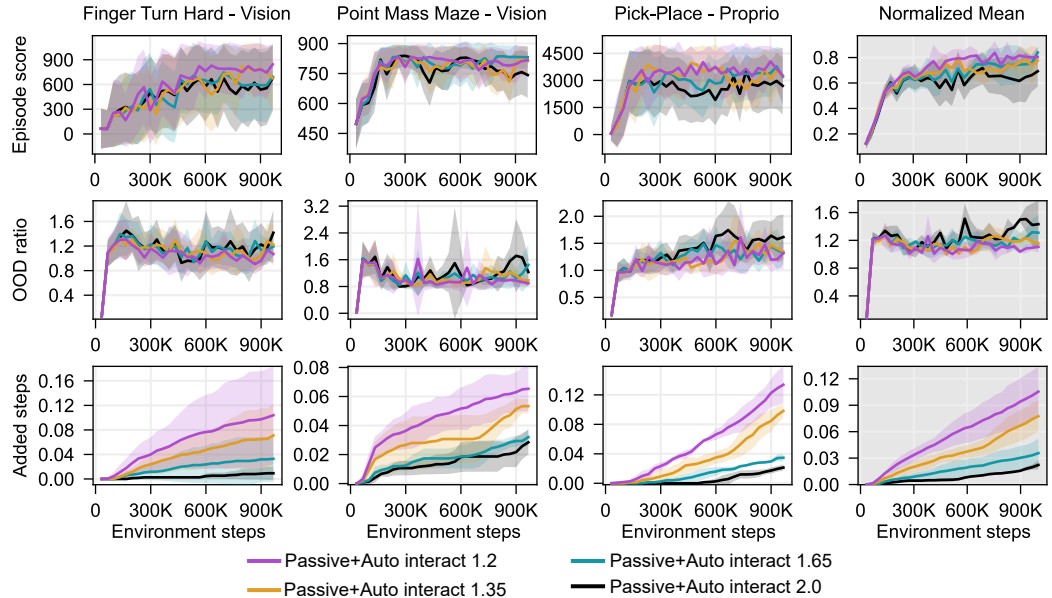

Figure S1: **Ablation studies on threshold value for adaptive Passive agents.** We test four threshold values: 2.0, 1.65, 1.35, and 1.2 in three tasks. The last column shows a normalized mean across tasks. The number of added steps in the third row is shown as a percentage of the original replay buffer size.

### A.1.6 Ablation Studies

We test different threshold values used in adaptive Passive agents for autonomously adding self-generated interaction data. In Fig. S14, we observe that the majority OOD ratio in Active agents reaches below 2.0 during training. Therefore, we begin with an upper bound threshold value of 2.0 and test four values: 2.0, 1.65, 1.35, and 1.2. It is important to note that this upper bound serves solely as a reference point for initiating the ablation studies and does not imply any dependence of the OOD_ratio on the performance of the Active agent. In Fig. S1, we show that although a lower threshold value (e.g. 1.2) could bring more self-generated data (about 10% average) to the replay buffer, the improvement in performance is not significant compared to other higher values. However, a high threshold value (e.g. 2.0 or 1.65) makes the training process less stable, as shown in the relatively low normalized mean score and an increasing tendency of OOD ratio from step 800K, compared to lower threshold values. But generally, the sensitivity of this threshold value to performance is low. One can set a low threshold value if the training budget allows. In the main experiments, we choose a middle threshold value of 1.35, which balances the number of added interaction data and stable performance.

### A.2 Supplementary of DreamerV3

The computation of each component in the world model loss:

$$
\begin{aligned}
\mathcal{L}_{\text{pred}}(\phi) &\doteq -\ln p_\phi(x_t \mid z_t, h_t) - \ln p_\phi(r_t \mid z_t, h_t) - \ln p_\phi(c_t \mid z_t, h_t) \\
\mathcal{L}_{\text{dyn}}(\phi) &\doteq \max\big(1, \text{KL}\big[\,\text{sg}(q_\phi(z_t \mid h_t, x_t)) \;\|\; p_\phi(\hat{z}_t \mid h_t)\,\big]\big) \\
\mathcal{L}_{\text{rep}}(\phi) &\doteq \max\big(1, \text{KL}\big[\, q_\phi(z_t \mid h_t, x_t) \;\|\; \text{sg}(p_\phi(\hat{z}_t \mid h_t))\big]\big)
\end{aligned}
\tag{S1}
$$

### A.3 Additional Metrics

**Policy input reconstruction loss** We train an autoencoder functioning as an OOD detector for the policy inputs. The autoencoder is optimized to minimize the negative log-likelihood (Eq. S2) to reconstruct the policy input. Novel policy inputs, that may compromise the quality of output actions, can be detected using the Mean Squared Error (MSE) reconstruction loss. A higher MSE indicates that the input is likely novel or anomalous, suggesting the input differs significantly from the training

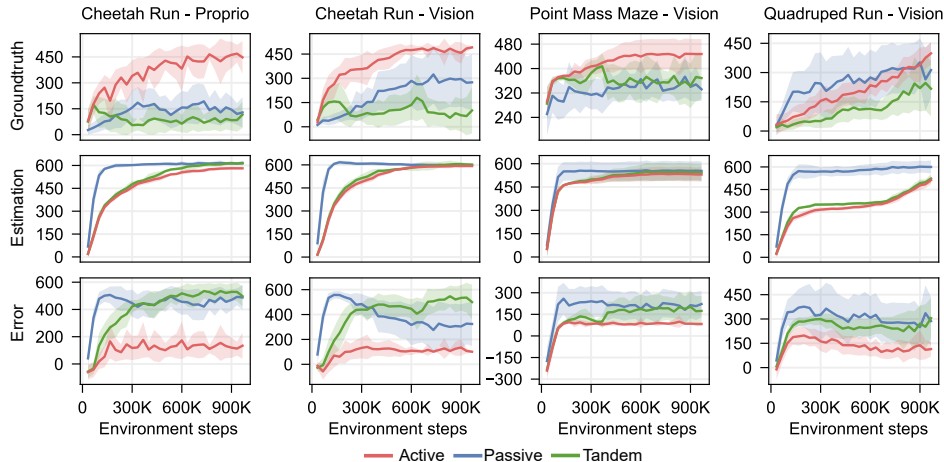

Figure S2: **Value function estimation of each agent.** The value function $V(s)$ is calculated on the initial state of each agent's trajectory, which should reflect the actual discounted rewards accumulated across the entire trajectory. The ground truth value is computed using Monte Carlo estimation from one sample trajectory. The error is computed by subtracting the ground truth value from the estimated value.

distribution and could lead to an unreliable policy action.

$$\mathcal{L}_{\text{recon}}(\phi) \doteq -\ln p_\phi(z_t, h_t \mid \text{encoder}(z_t, h_t)) \tag{S2}$$

**Value function** The expected discounted return—the cumulative sum of future rewards, as shown in Eq. (1).

The additional metrics are calculated as follows unless specified otherwise: (1) Every 5K environment steps, we roll out the agent's policy for a total of 4 episodes. (2) We compute the policy input reconstruction loss across the 4 episodes. For the value function, we calculate it at the initial state of each episode trajectory and then average these values across the 4 episodes.

### A.4 Discrepancy between Imagination and Real Rollouts

As outlined in Sec. 2.1, the agent's policy utilizes an actor-critic framework, with the critic predicting the value function $V(s)$ for each given state. Since the critic is trained in the imagination of the world model and will subsequently be used to train the actor, it is essential that its value estimates accurately reflect the agent's real rollout conditions. If the actual rollout performs poorly, a correctly low-value estimate from the critic can guide the actor's updates in a direction that improves performance. However, in Fig. S2, we show that both Passive and Tandem agents consistently wrongly predict their value functions, assigning high values even when their actual trajectories yield low rewards. Throughout training, the value function estimation error for these offline agents remains significantly higher than that of the Active agent, showing consistent statistical differences across time scales. This finding highlights that, without the self-correction mechanism, offline agents exhibit a substantial discrepancy between imagined and real rollouts, evident in the differences between estimated and ground truth value functions. This misalignment can lead to suboptimal actor updates, ultimately resulting in unstable or degraded performance.

### A.5 Per-step Analysis of Performance Degradation

#### A.5.1 Impact of Novel States during Evaluation

**Novel states disrupt world model output and therefore agent performance during evaluation.** After the agent enters into novel states, the world model will output inaccurate estimations and latent embeddings. Since the policy network relies on these inaccurate latent states as input, this can start the catastrophic cycle where each compromised action leads to further novel states and additional inaccuracies until the episode ends or the agent accidentally re-enters into a familiar state. In Fig. S3, we provide for two test times trajectories the reward, world model loss, and policy reconstruction loss across two tasks. A low task reward is typically accompanied by a high world model loss. A high

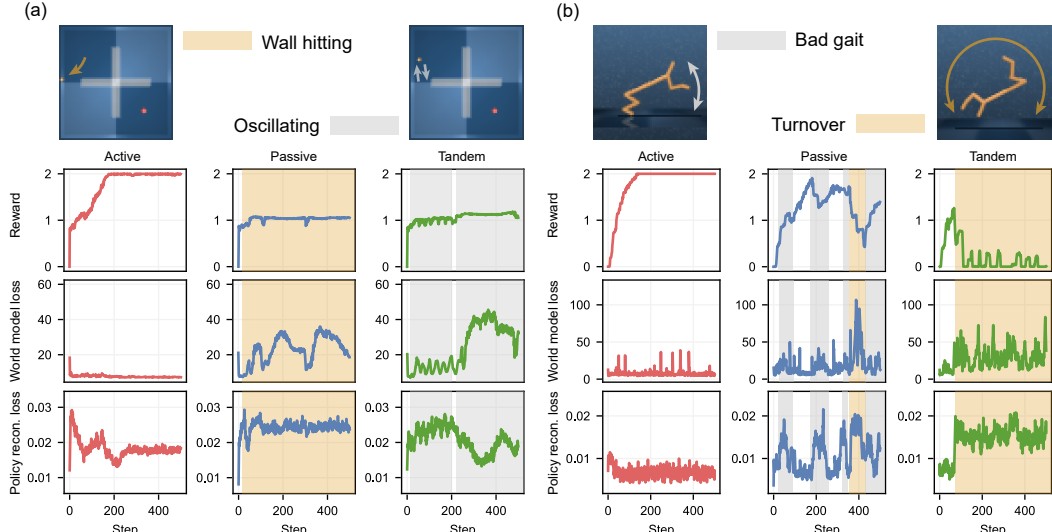

Figure S3: **Stepwise analysis within a single test episode** of the Point Mass Maze - Vision and Cheetah Run - Vision tasks from DMC. The plots show the progression of reward, world model loss, and policy input reconstruction loss at each step as the agent executes actions given by its own policy. Timesteps, where agents exhibit abnormal behavior, are highlighted with yellow and grey regions. Each episode consists of 500 steps, with the environments initialized identically across agents. The agents are the fully trained version after 1M environment steps.

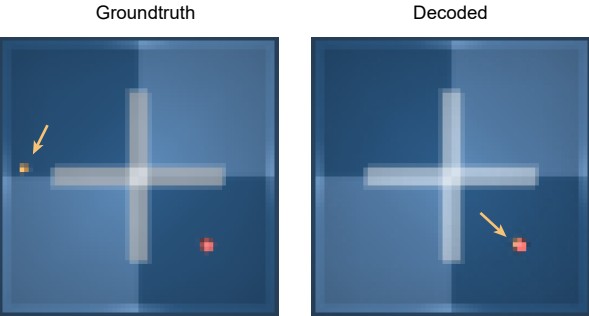

Figure S4: **World model misinterprets the novel states.** In the decoded image (step 324 in Fig. S3) from the world model of the Tandem agent in task Point Mass Maze - Vision, the ball appears at the goal position while in the ground truth observation, it is actually in a novel region to the world model.

world model loss typically indicates a high policy input reconstruction loss, meaning the policy is unfamiliar with such inputs, leading to compromised actions. For task (a) *Point Mass Maze - Vision*, the agent never returns to a familiar region once it hits a wall. Similarly, in the task (b) *Cheetah Run - Vision*, the Passive and Tandem agents turning over also reaches such OOD states; however, the Passive agent can recover from the OOD state - the task setting and the environment dynamics allow to recover more easily, temporarily ending the catastrophic cycle. This is evident from the intervals of successful actions between failure periods in the Passive agents.

**World model can sometimes hallucinate and mislead policy in novel states.** We observe unexpected instances where the policy input reconstruction loss remains low even when the world model loss is high, as seen between timestep 300 and 400 in the Tandem agent of the *Point Mass Maze - Vision* task in Fig. S3. With closer examination in Fig. S4, the decoded image by the world model shows the agent has already reached the target position while, in fact, it is still far away from the target. It indicates that the world model hallucinates in the novel states and produces an incorrect mapping of the latent state during that period. In this case, the latent state is no longer novel to the policy, which makes the policy input reconstruction loss ineffective in detecting performance degradation and misleads the policy to output inadequate actions.

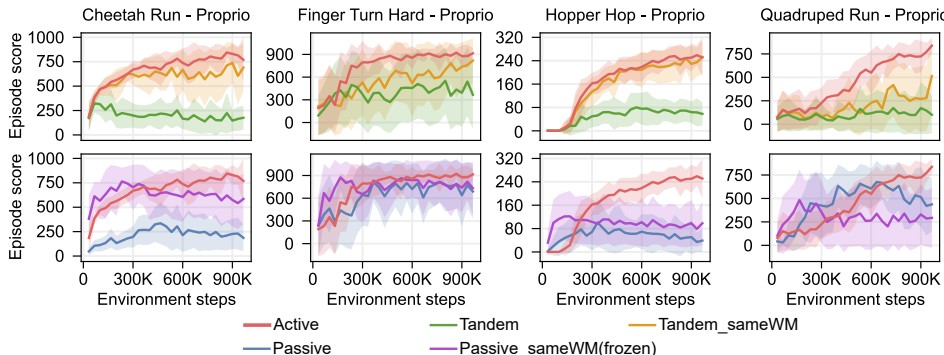

Figure S5: **Performance comparison when keeping an equivalent world model in Passive or Tandem agents to the one of the Active agent** throughout training. Despite utilizing the same world model during training, performance degradation still occurs, albeit to varying degrees.

## A.6 BOTH WORLD MODEL AND POLICY AFFECT PERFORMANCE DEGRADATION

To investigate which one, world model or policy, plays the most important role in causing the performance degradation, we carry out a more controlled experiment in Fig. S5. In this setup, the Tandem agent's world model is synchronized with that of the Active agent, replicating its neural network weights precisely at each training step. This variant, referred to as Tandem_sameWM in Fig. S5, differs from the Active agent only in the initialization of the policy network. For Passive agents, we initialize with the final world model from their Active counterpart, then freeze the world model for the remainder of training. This variant is named Passive_sameWM(frozen) in Fig. S5. After isolating the effect of different world models on performance degradation, we observe that the degradation still persists even when using an identical world model to the Active agent. However, the extent of degradation varies across tasks. In tasks such as *Hopper Hop - Proprio*, the performance degradation of the Tandem_sameWM agent is minimal, while it remains significant in others like *Quadruped - Proprio*. A similar trend is observed with the Passive_sameWM(frozen) agents. These findings suggest that deviations in both the world model and policy from the Active agent contribute to performance degradation, with their relative impacts depending on the specific task. In the Passive_sameWM(frozen) agent for the *Quadruped - Proprio* task, we observe an interesting case where performance degradation is even more severe than in the original Passive agent. This result further highlights that, without the self-correction mechanism, relying on a well-trained world model alone is insufficient for achieving good task performance in a different agent.

## A.7 DETAILED RESULTS OF CONSIDERATIONS IN PRACTICAL APPLICATIONS

**Advantage of training agents offline**    Although the performance degradation caused by the OOD issue is prominent in Passive agents, they show potential for faster convergence and more efficient training, as seen in tasks like *Quadruped Run - Vision* and *Pick-Place - Proprio* in Fig. 3. This is because Passive agents have access to high-quality trajectories from the beginning, while Active agents must wait until later in training to encounter those trajectories. We validate this hypothesis in Fig. S8, where Passive agents trained on suboptimal data generally perform worse than those trained on mixed data. It indicates that mixing expert trajectories into suboptimal data helps the performance, which matches the case between the Active (suboptimal data) vs. Passive (mixed data) agent in the early training stage. Therefore, addressing the OOD issue in Passive agents is crucial, as solving it could unlock the potential for highly efficient agent training. However, we do not observe such advantages in Tandem agents.

**Different model initialization**    In this section, we answer the question whether the model initialization affects the performance degradation. In particular, if we initialize the world model and policy network of a Passive agent using the same seed as the Active one, will the performance differ from the independently initialized Passive agent? In Fig. S6, we show that no significant difference in the task performance can be observed with initialization seeds among Passive agents. We also investigate the sensitivity of task performance to the initialization of weights in model networks of Tandem agents. By mixing weights of the identically initialized networks as the Active and those of an

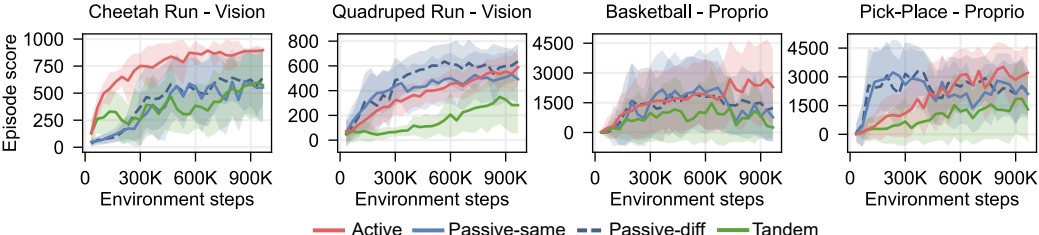

Figure S6: **Model initialization matters not in Passive agents.** Performance comparison when initializing the world model and policy network of Passive agents with the same and different seed w.r.t. the Active agents.

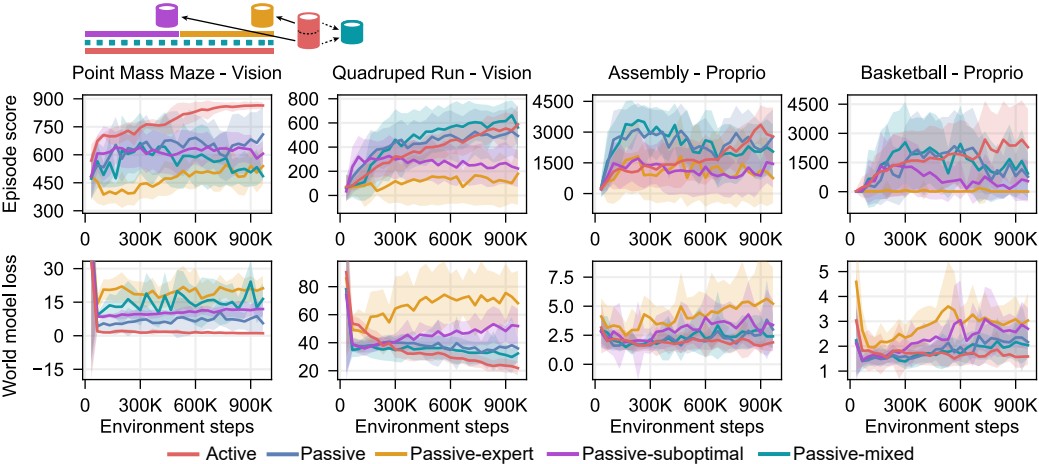

Figure S8: **Performance comparison when training Passive agents on different halves of the replay buffer from the Active.** We split the replay buffer (red bucket) at the 500K environment steps, as shown in the schematic illustration on the Point Mass Maze - Vision. The first half (purple bucket) represents the suboptimal data, while the second half (yellow bucket) mainly contains high-reward expert data. Therefore, Passive-expert, Passive-suboptimal, and Passive-mixed have a halved replay buffer compared to the normal Passive agent. The replay buffer of the mixed agent (turquoise bucket) is uniformly sampled from the whole replay buffer.

independent initialization with different ratios $\alpha$, it allows us to observe whether a tiny difference in the initialization will cause a big difference in task performance.

$$w \doteq (1 - \alpha) \cdot w_{\text{Active}} + \alpha \cdot w_{\text{Tandem}} \tag{S3}$$

In Fig. S7, we observe that even a small deviation from the weights of the Active agent eventually causes a large difference in task performance when training on the identical sequence of training batches each training step.

**World model overfitting on expert dataset** Another popular practice to facilitate training a capable agent is to train the agent on an expert dataset (Kumar et al., 2022). However, in Fig. S8, we find that training on expert data leads to an even worse performance degradation in Passive agents. It is also indicated by the high world model loss with a growing tendency. However, according to the performance of Passive-mixed agents, mixing expert data with suboptimal trajectories can help mitigate this issue. The expert dataset primarily consists of monotonic task-solving trajectories, which implies extremely limited state-space coverage. Incorporating suboptimal data expands this coverage during training and reduces the OOD issue during policy rollouts in evaluation. This highlights the importance of broad state-space coverage during training and the need to include

Figure S7: **Performance comparison of the world model and policy network of Tandem agents initialized with mixed weights.** Results shown for different $\alpha$ values (indicated in run name) as defined in Eq. (S3). Results for one seed.

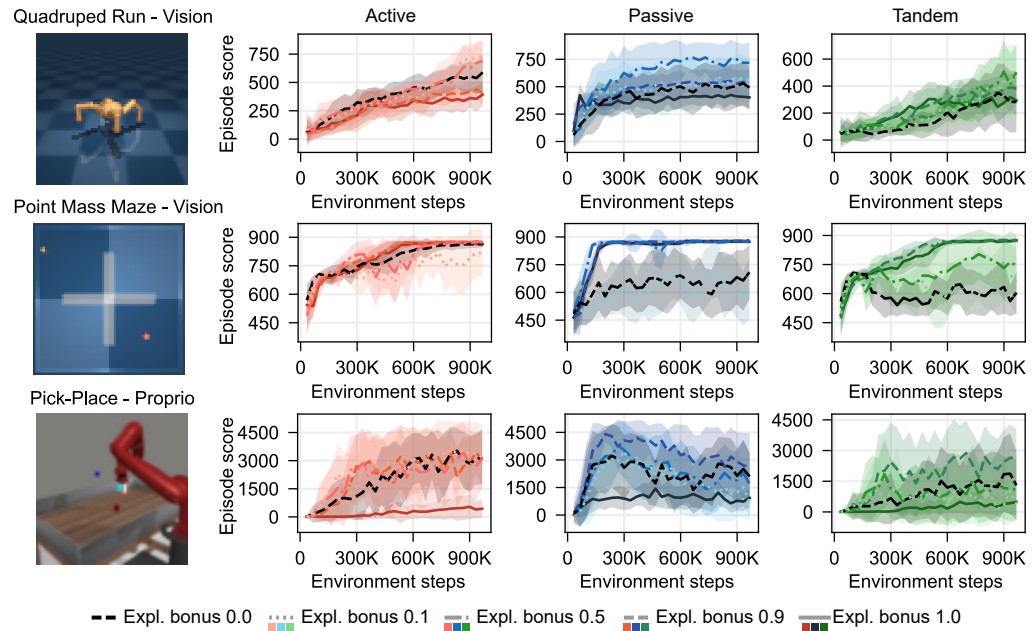

Figure S9: **Different task has different optimal exploration bonus values.** Performance comparison when assigning different exploration bonuses $w_{\text{expl}}$ in the reward function. The black dashed lines represent pure task-oriented policy without any exploration bonus.

exploration-equivalent data to ensure a capable agent. This finding matches results from previous research (Gulcehre et al., 2021; Mediratta et al., 2024; Suau et al., 2023).

**World model overfitting on low-dimensional inputs** In the *Basketball - Proprio* and *Pick-Place - Proprio* tasks, the performance of the Passive agent declines as the world model loss increases in the second half of the training process. A similar issue is observed in proprioceptive versions of DMC tasks in Appendix A.8.2. It indicates that the world model begins to overfit on the fixed data distribution in the replay buffer, given that the Passive agent is not allowed to add its own interaction data and cannot change the data distribution progressively in the same way as the Active agent. This tendency is pronounced in the proprioceptive version because of a lower input dimension for the world model than image-based observation, more prone to overfitting.

## A.8 COMPLETE RESULTS

### A.8.1 RESULTS OF AGENTS WITH DIFFERENT EXPLORATION BONUS

In Fig. S9, we show all three analyzed tasks with comparison among different exploration bonus values. The optimal exploration bonus $w_{\text{expl}}$ is 0.5 for task *Quadruped Run - Vision*, 0.9 for tasks *Point Mass Maze - Vision* and *Pick-Place - Proprio*.

### A.8.2 RESULTS OF TASK-ORIENTED AGENTS

In Fig. S10 and Fig. S11, we show the complete results in 31 tasks corresponding to the discussion in Sec. 3.4 and Sec. 3.5. The Passive agent initialized using the same seed for the world model and policy network as the Active agent is marked with a suffix "-same", while the different model initialization is marked with "-diff".

### A.8.3 RESULTS OF ADDING SELF-GENERATED DATA

In Fig. S12, Fig. S13, and Fig. S14, we show the complete results in 31 tasks, where we allow the Passive agents utilize the self-generated data from environmental interaction, corresponding to the discussion in Sec. 4.2. In Tab. S2, we show how many self-generated data is added to the replay buffer by Passive+Auto interact agents. The percentage is calculated using the number of additionally added

steps divided by the total number of steps in the original replay buffer. In Fig. S15, we also show that our adaptive agent **Passive+Auto interact** can converge fast and require minimal interaction data to recover the performance.

Table S2: Percentage of added self-generated data by Passive+Auto interact agents

| Task | Percentage (%) | Task | Percentage (%) |
|---|---|---|---|
| cheetah_run-proprio | 10.44% | walker_walk-proprio | 18.27% |
| cheetah_run-vision | 6.53% | walker_walk-vision | 7.87% |
| cup_catch-proprio | 0.67% | assembly-proprio | 8.04% |
| cup_catch-vision | 9.47% | basketball-proprio | 7.16% |
| finger_turn_hard-proprio | 2.53% | button-press-proprio | 4.04% |
| finger_turn_hard-vision | 3.47% | lever-pull-proprio | 1.20% |
| hopper_hop-proprio | 4.31% | peg-insert-side-proprio | 2.31% |
| hopper_hop-vision | 4.00% | pick-place-proprio | 9.82% |
| humanoid_walk-proprio | 17.78% | soccer-proprio | 14.93% |
| humanoid_walk-vision | 3.60% | window-open-proprio | 1.47% |
| point_mass_maze-proprio | 0.00% | asterix-vision | 2.68% |
| point_mass_maze-vision | 4.62% | breakout-vision | 1.86% |
| quadruped_run-proprio | 2.53% | freeway-vision | 0.00% |
| quadruped_run-vision | 2.93% | seaquest-vision | 0.07% |
| reacher_hard-proprio | 2.27% | spaceinvaders-vision | 0.47% |
| reacher_hard-vision | 20.31% | **Average** | **5.67%** |

### A.8.4 RESULTS OF EXPLORATIVE AGENTS

In Fig. S16 and Fig. S17, we show the complete results in 31 tasks using agents with pure exploration rewards, corresponding to the discussion in Sec. 4.1. The Passive agent initialized using the same seed for the world model and policy network as the Active agent is marked with a suffix "-same", while the different model initialization is marked with "-diff".

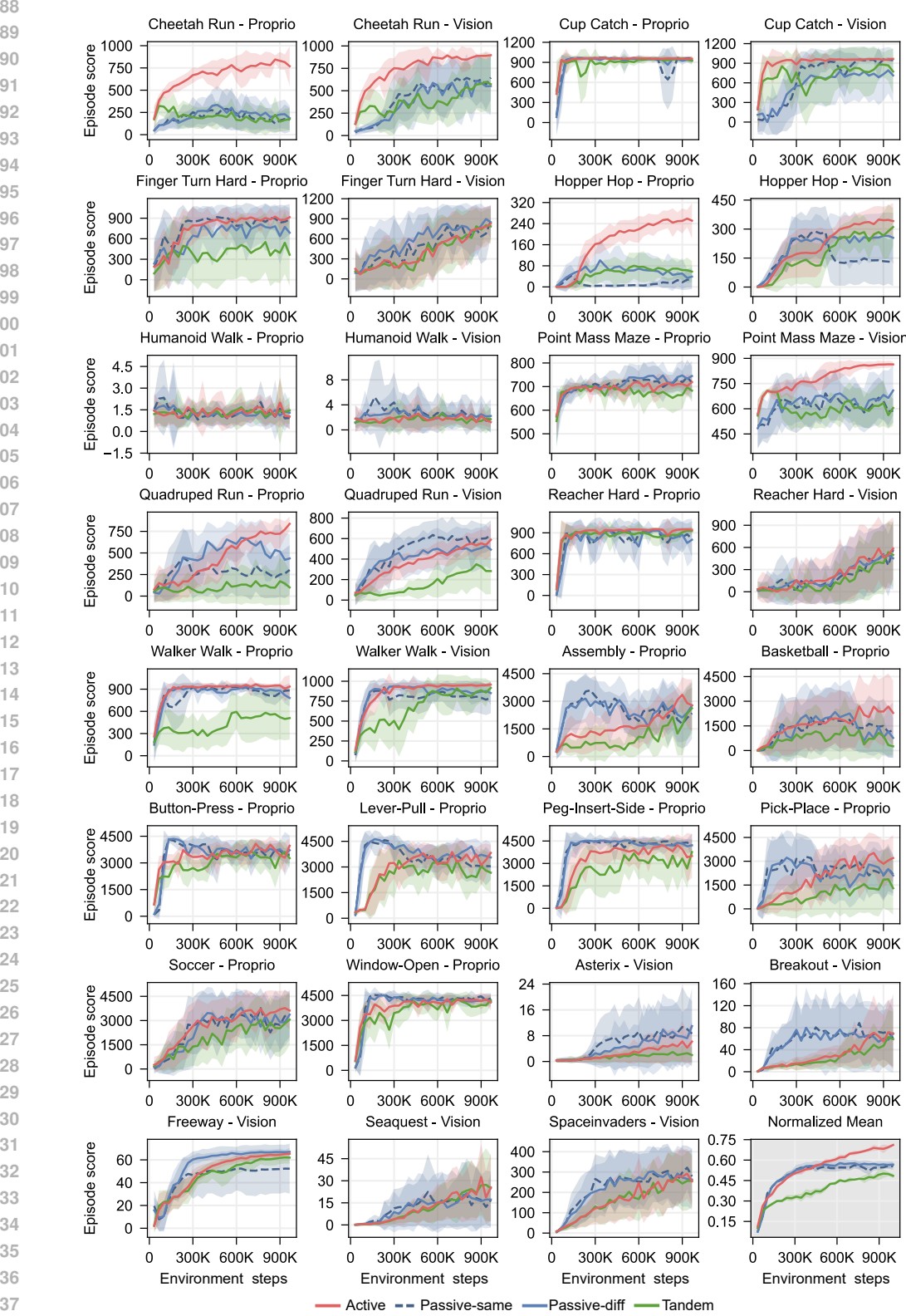

Figure S10: **Episode score of 31 tasks.** The first 18 tasks are from DMC, the subsequent 8 tasks are from Metaworld, and the last 5 are from the MinAtar domain. We also output a normalized mean score across tasks. The Passive-same is Passive agents initialized identically as the Active agents while Passive-diff is independently initialized.

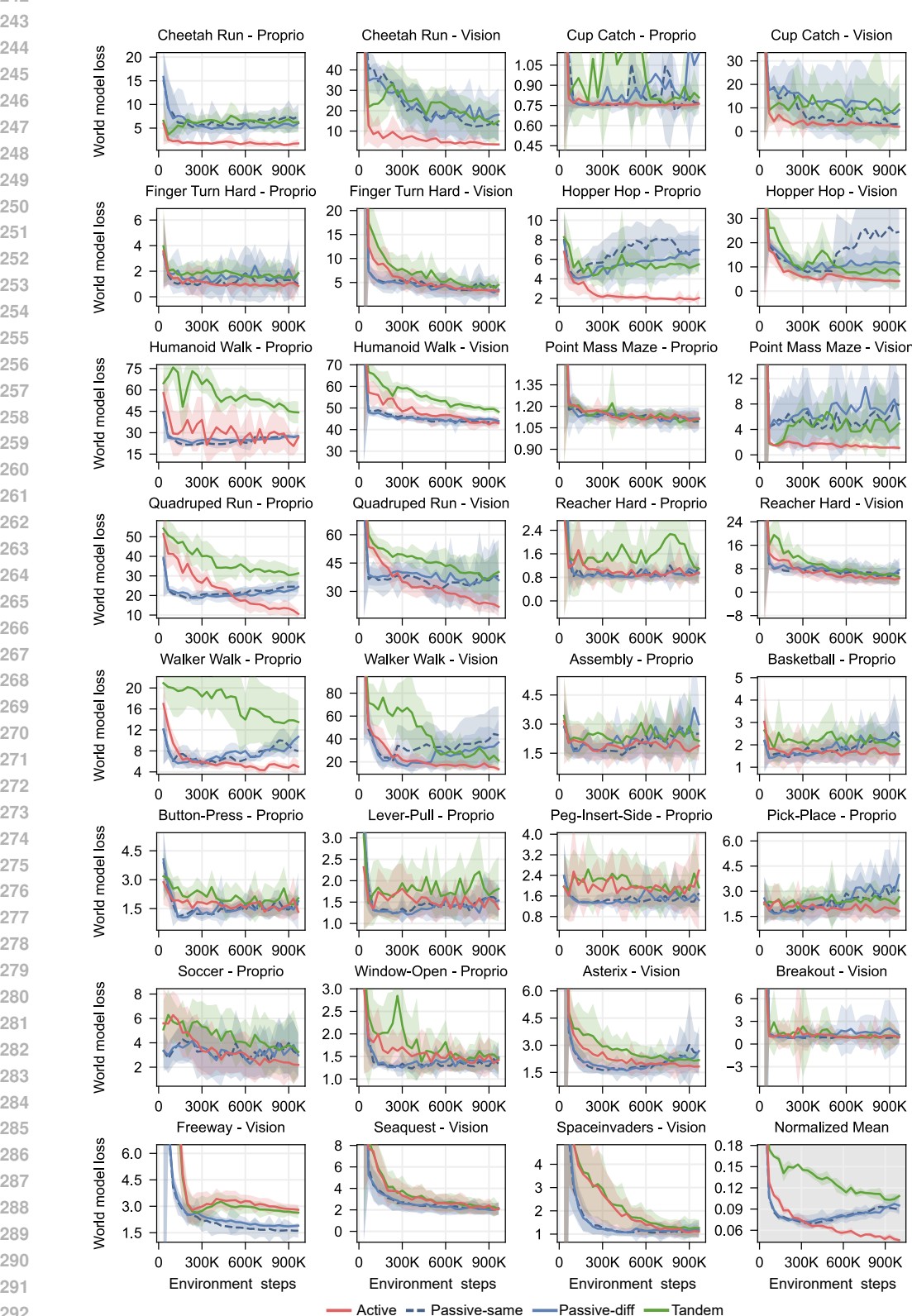

Figure S11: **World model loss of 31 tasks.** In the last subplot, we show an additional normalized mean result across tasks.

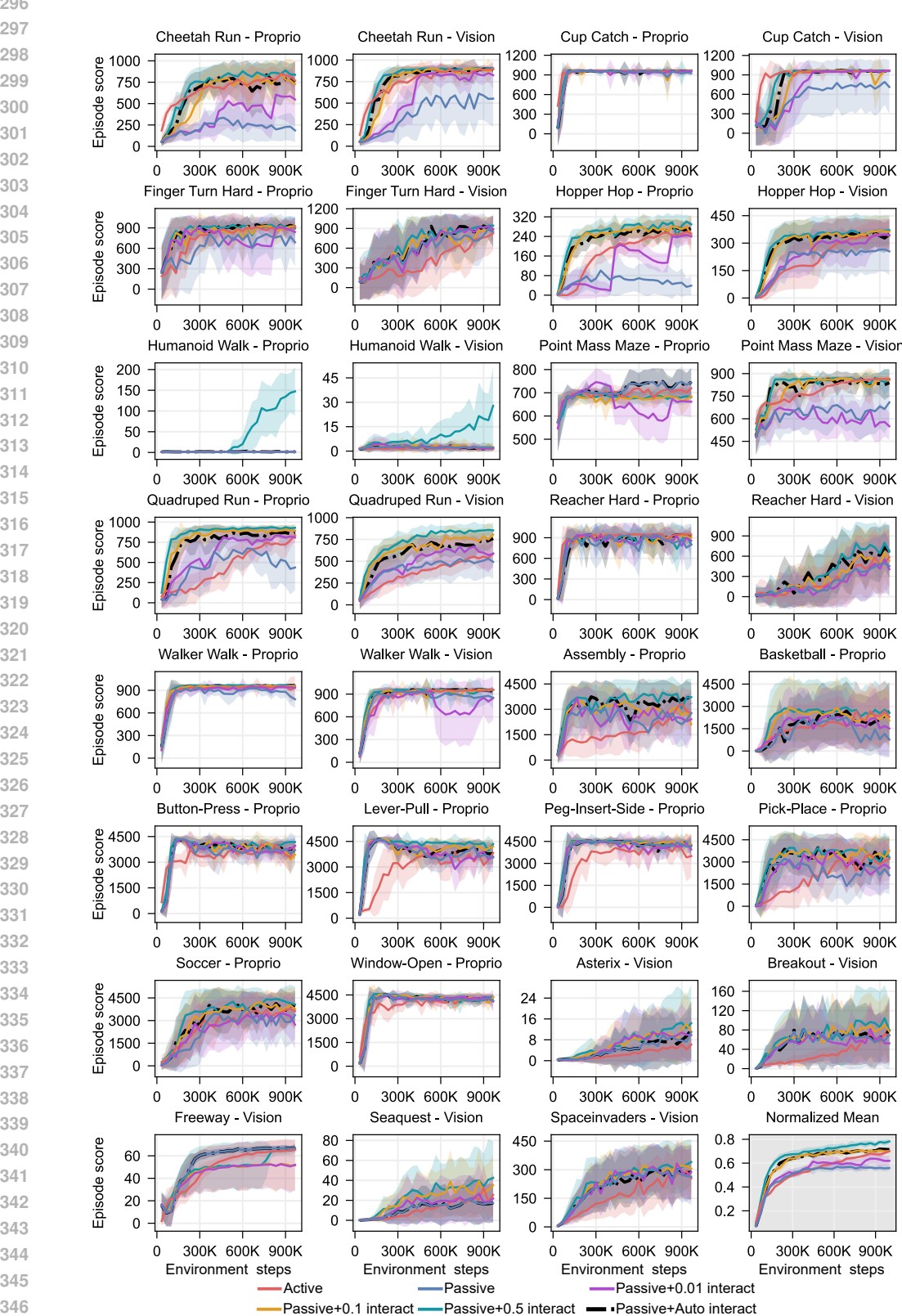

Figure S12: **Episode score of 31 tasks.** In the last subplot, we show an additional normalized mean result across tasks.

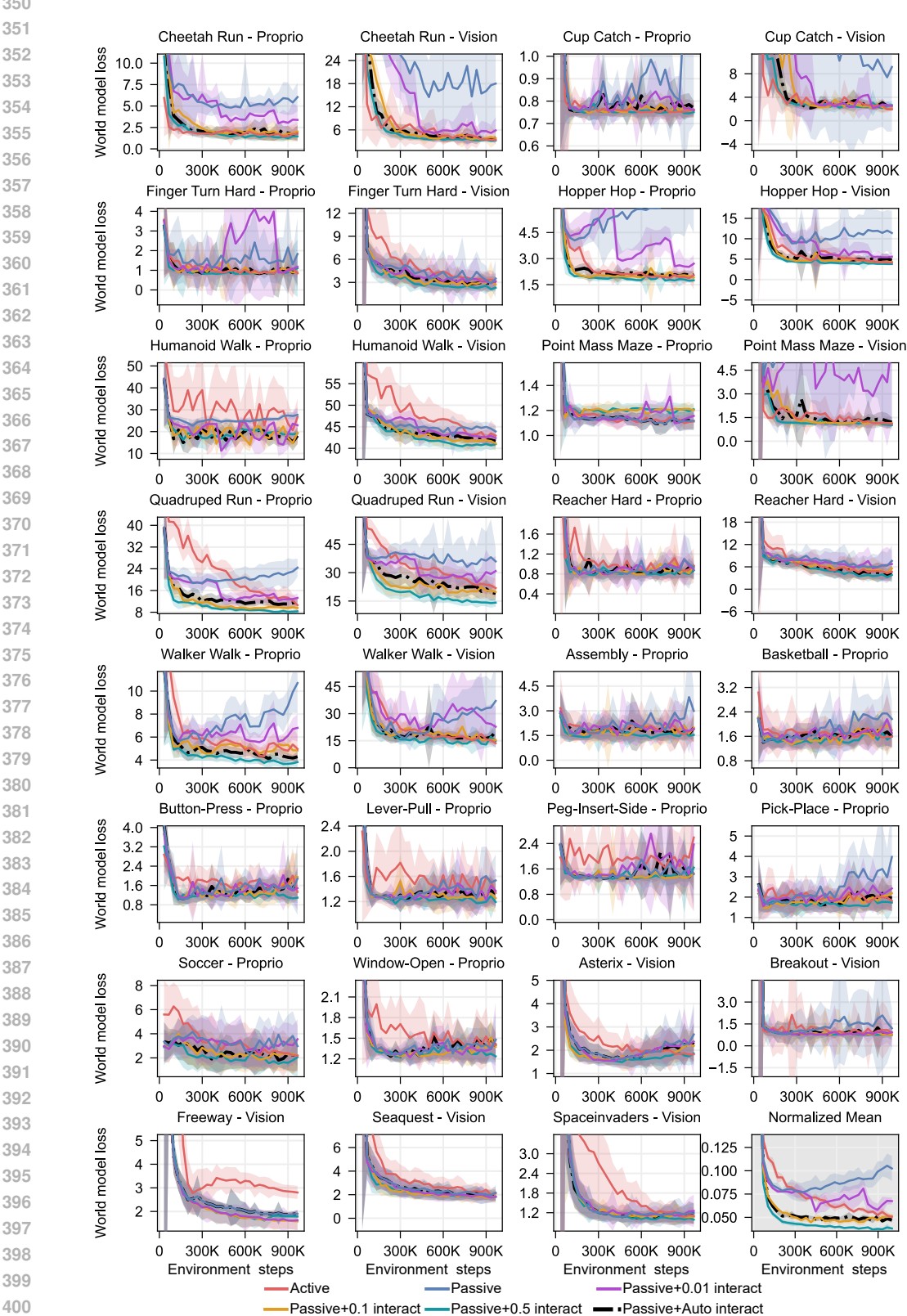

Figure S13: **World model loss of 31 tasks.** In the last subplot, we show an additional normalized mean result across tasks.

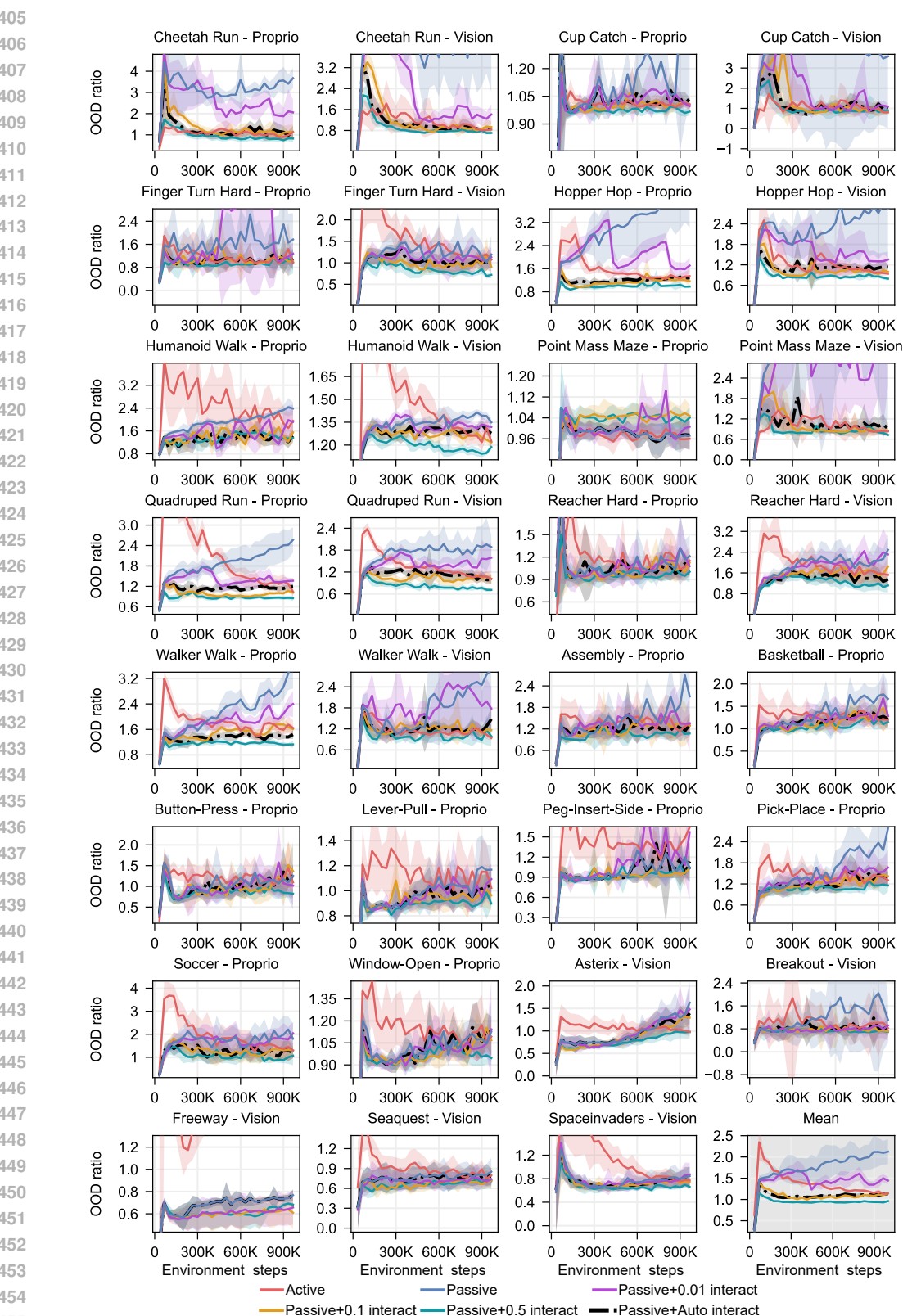

Figure S14: **OOD ratio of 31 tasks.** In the last subplot, we show an additional mean result across tasks.

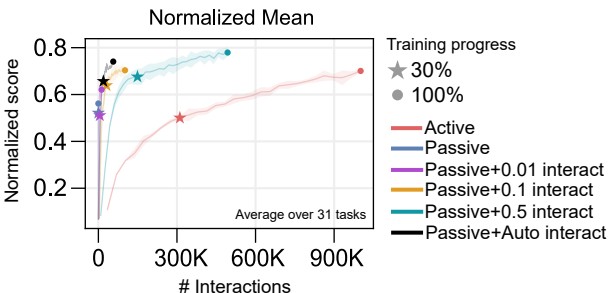

Figure S15: **Performance comparison between different Passive agents allowed environment interaction.** The y-axis is the average normalized episode score across 31 tasks. The x-axis shows how many self-generated interaction data are added to the replay buffer. Generally, an agent with markers closest to the top left corner is the best, having the fastest convergence speed and highest score, and requires minimal self-generated interaction data.

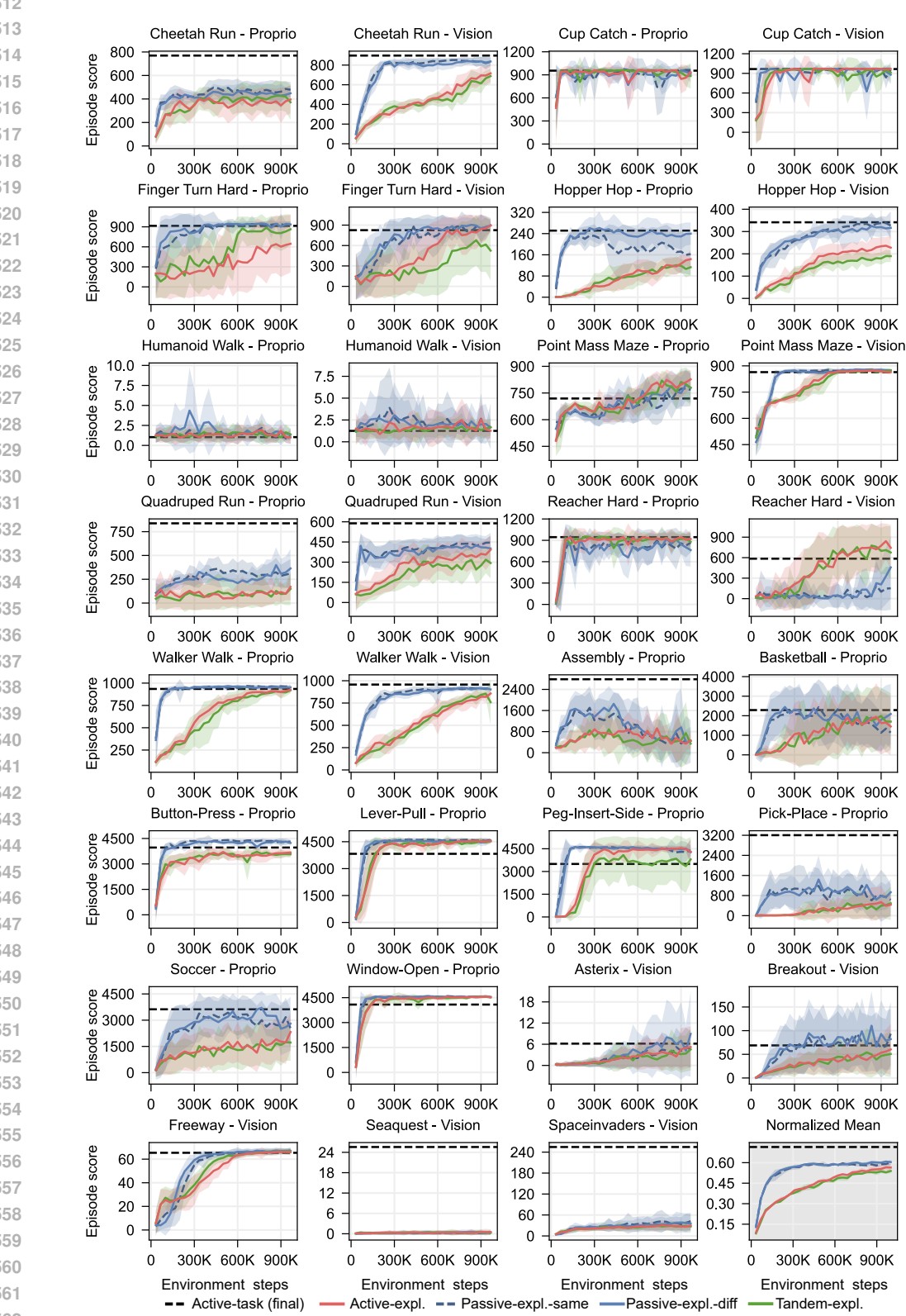

Figure S16: **Episode score of 31 tasks using agents with pure exploration rewards.** We also show the final performance of a task-oriented Active agent as the baseline in black dashed horizontal lines. In the last subplot, we show an additional normalized mean result across tasks.

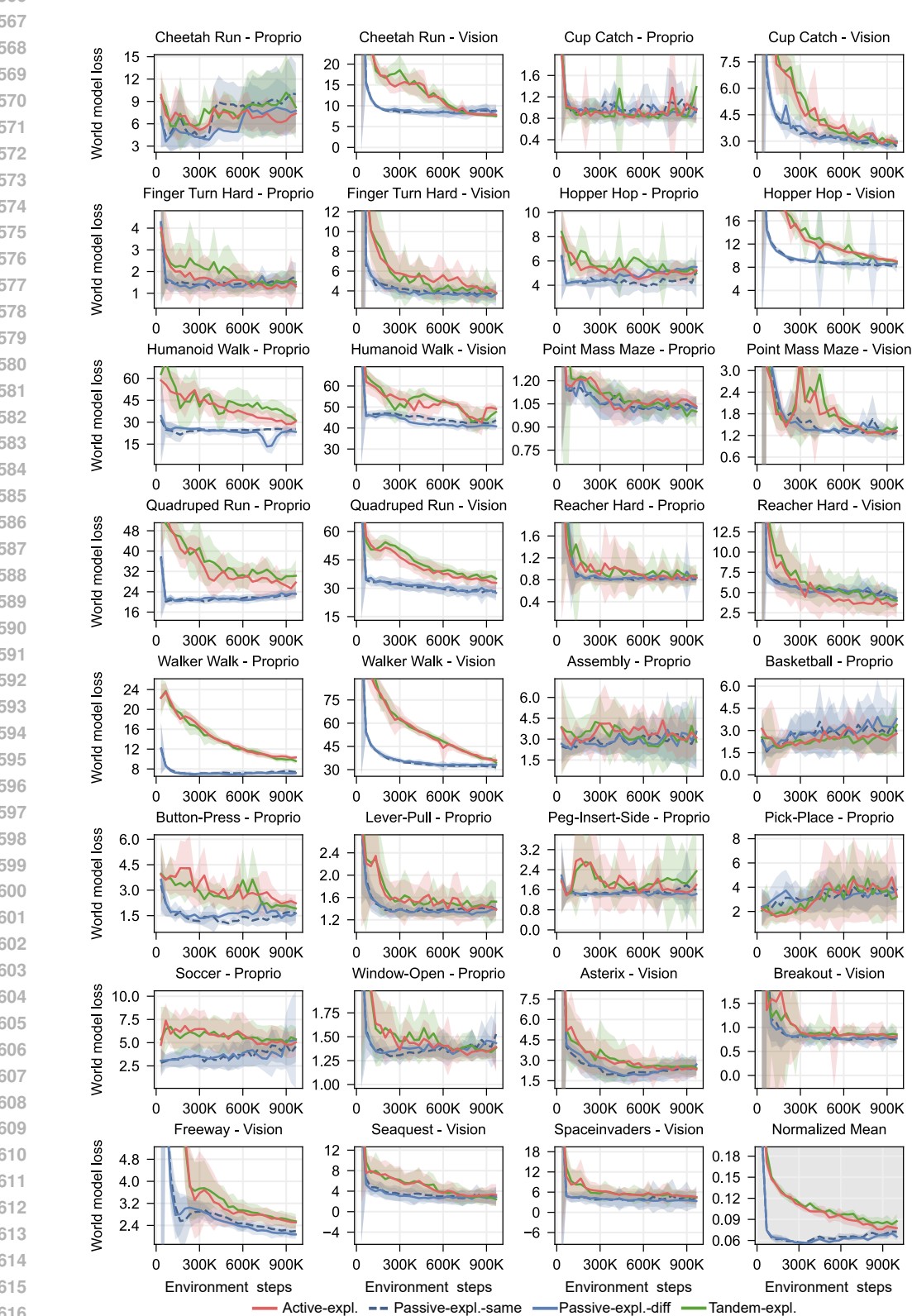

Figure S17: **World model loss of 31 tasks using agents with pure exploration rewards.** In the last subplot, we show an additional normalized mean result across tasks.

