# OpenReview forum: "Offline vs. Online Learning in Model-based RL: Lessons for Data Collection Strategies"
_ICLR.cc/2025/Conference — Submitted to ICLR 2025_

### Official Review · Reviewer_tze1 · 2024-10-31

**Soundness:** 2
**Presentation:** 3
**Contribution:** 2
**Rating:** 6
**Confidence:** 3

**Summary:**

This paper focuses on the data collection strategy for model-based RL. It is principally concerned with three types of agent: 'Active', which is allowed to interact with the online environment to update its world model (a bit like Dreamer); 'Passive', which samples random data from the replay buffer of the 'Active' agent; and 'Tandem', which receives identical data to the 'Active' agent but has a different initialisation. They show that these three agents have a performance disparity, and subsequently consider some different sampling techniques to improve the fidelity of the world model or the agent performance (such as including exploration bonuses). There is an extensive appendix, largely comprised of additional results.

**Strengths:**

I found this to be, besides certain confusing sentences, a generally well written and presented paper. That said, I feel it is significantly lacking in contribution. Much of what it says is seemingly obvious or feels like shifting goalposts to fit a narrative.

I provide a list of my perceived strengths of this paper below:
- As stated, this was a generally well written and formatted paper.
- The paper is extensive in its experimentation, covering a broad range of environments and settings.
- The paper is well referenced, and provides good coverage of related literature. In weaknesses I provide a few papers which I believe are missing from the literature review but, generally, the authors have done well to contextualise against prior work.
- I think the paper is well structured, following a reasonable trajectory of 'What we explore -> Some analysis -> Mitigation strategies'
- Defining notation used throughout the paper early, and remaining consistent with plotting colours etc. even when more or less baselines are included, helped make results more interpretable. This holds even to figure 1, which is a nice touch.
- I think the motivation of the paper is good - too often there is a focus on proposing new methods in research, so I appreciate this analytical approach to understand the current deficiencies in model based RL. Asking the question of 'what is the best data collection strategy for 'MBRL' is a good one, though I explain some of my hesitations about this in the weaknesses section.
- Though I would question if **all** the experiments are needed, especially due to their limited reference in text, the authors provide the necessary information for both reproducibility and compute requirements.

**Weaknesses:**

My principle concern with this work, and what limits me from increasing my score, focuses on its significance; I think there are interesting questions about if this is potentially misguided, which I get into below, but overall my main assessment is that a lot of these findings are quite trivial. For instance, it is well known that training a world model on purely expert trajectories will lead to subpar policies due to overestimation bias in OOD states, and in that situation an imitation learning approach may be more expected. Similar, it should not be a surprise that introducing additional data to the agent via online sampling will lead to better performance.

Below, I cover more specific feedback. That said, I think the researchers behind this work are asking an important question and have clearly put a lot of effort into this project. I am keen to discuss this with the authors in more depth as, while I hold strong conviction after reading the paper, I am open to being persuaded that I have misjudged the work. I apologise that the review has ended up longer than expected, though I think many of the points have a degree of overlap and thus am happy for the authors to respond more holistically to the points I raise rather than going through bullet point by bullet point.

- I found lines 19-23 in the abstract very confusing. I think part of my view on the misguidance in this work is the flippant use of 'offline' and 'online' to refer to data collection. For instance, in the abstract it says the below, which seems to suggest that the problem with  offline learning is that it collects data online:
> We identify a key challenge behind performance degradation of offline agents: encountering Out-of-Distribution states at test time. This issue arises because the data collected online primarily benefits the online agent by learning from its own mistakes, but it leaves many states unvisited. As a result, the offline agent suffers from insufficient coverage of the state space.
- Another point I feel is not addressed is that an offline dataset is generally colelcted prior to policy learning and, as such, is not designed. The paper focuses on finding better ways to shape offline datasets (eg line 28 in the abstract, adding exploration data), ignoring the fact that an offline dataset is presumably pre-collected and there is not ready access to the online environment to collect more data.
- I think it is misinformed to say current efforts predominantly focus on expert data (line 29) when arguably the most common offline benchmark, D4RL, has random, medium and mixed policy datasets in addition to their robotics data.
- I think there are some missing reference examples; most notably, World models by Ha & Schmidhuber is a seminal MBRL paper. I think a relevant recent paper is The Edge-of-Reach Problem in Offline Model-Based Reinforcement Learning, Sims et al. (2024).
- I am a bit unclear on Figure 1. We are focused on model-based RL here, but there seems to be no demonstration of the world model in figure 1 - just an agent.
- The explanation of the active agent is very unclear.
- Equation 5 introduces a policy input reconstruction loss, but this does not seem to be used elsewhere in the paper.
- On line 217, it is worth pointing out that Dreamer V3 has the online data collection discussed here. I am not quite sure what the argument is.
- I am unsure what is means in line 234 onwards:
> Their mistakes are catastrophic, as their world models and thus their policies have their unique fallacies that cannot be corrected by the task-oriented data with limited state space coverage, which is collected by and catered to the Active agent.
- There is a lot of discussion about how the passive agent experiences worse performance than the active agent in section 3.4, but again this seems quite trivial; the agent is not able to interact with the real-world so effectively does not receive any online feedback about its overestimation bias. Just because it is trained on the same data as the active agent, it is obvious that this lack of correction would harm performance; the reason many don't sample online during training isn't because they think it won't help, but because it is impossible or impractical.
-I'm a bit unsure about what is meant by line 300. Again, there is little insight in the fact that the replay buffer trajectories will have a lower reconstruction loss since they make up part of the training signal of the model; this is equivalent to having a higher supervised test loss compared to training.
> However, in our observation, the quality of imagined trajectories for all agents is poor with mean world model loss being more than 5 to 10 times higher compared to trajectories in the replay buffer due to accumulated error in open-loop multi-step predictions, as shown in Appendix A.4
- When considering potential mitigations, I again bring up what I believe to be misguidance in this paper. For instance, I think assuming we are rolling out a policy for collecting offline data to do offline RL with (4.1) can be potentially misguided - in this case we could just do online RL. Instead, I think an assumption in offline RL is that there is a precollected offline dataset that we then have to work with. Similarly, I think the fact that an exploration only dataset underperforms is expected, but I think this hinges on the fact that the dataset size is small.
- In section 4.2, it is suggested that offline data is 'often cheaply available', which seems in conflict with the statement in the introduction that 'collecting expert trajectories is expensive'. Again, it should be no surprise that allowing an agent to interact with the real world will increase performance.

I also raise some very minor points and typos below.
- Lots of the bibliography is incorrectly formatted, seemingly including the names of authors from different papers.
- There are a number of 'bulk references' (i.e., a large number of papers cited in a row) without explanation of what each of these papers are or their relevance to the statement. It would be good to disambiguate between these many references.
- In equation 1, $\pi$ is used on both sides of the equation which is unintuitive. The LHS should be labelled $\pi^*$
- In the explanation of Dreamer, $R_t$ is used to define the critic without definition.
- Pages 8 and 9 have a large amount of whitespace. It would look better to remove this via formatting.
- On line 514, there is a missing space after the full stop: 'best facilitates offline training.In'

**Questions:**

- I really don't understand how the Tandem agent with the same random seed and exact same data as the active agent could perform worse; a chaotic system is one where a small change in inputs causes large changes at output, but that is not the case here. Can you please elaborate how there could be any difference between the Tandem and Active agent please? Is the only difference here hardware determinism?
- In the discussion of pessimistic indication of performance degradation, I am a bit confused. Is the argument that model loss underestimates performance *degradation* (as in agents perform better than their model loss suggests) or that model loss underestimates performance (agents perform worse that their model loss suggests)?
- Is it not true that you have an offline collected RL dataset, and thus are unlikely to be able to curate this yourself?

---

> ### Author Response · Authors · 2024-11-23
>
> We thank the reviewer for providing detailed feedback on our paper.
>
> For **Weakness 1**, the ‘online’ and ‘offline’ data collection mean two different ways of how they get its training data:
>
> *online*: collect its on its own (e.g. rollout its own policy in env)
>
> *offline*: use someone else’s dataset (e.g. collected by another agents)
>
> In all our experiments, the Active, Passive, and Tandem agents use the same replay buffer; however, this buffer is online for the Active agent (because it collects this buffer itself) but offline for the Passive and Tandem agents. The problem with offline learning is that it does not collect data online; hence, it lacks the self-correction mechanism. We also update lines 20-23 in the Abstract to better fit our points.
>
> For **Weakness 2 & 11**, it’s true that if the data is already there and there is no access to online env, then the proposed strategies could be less meaningful. However, as we highlight in the General Response Q2, we are not proposing novel solutions, but sharing our exploration and understanding on the important dataset properties to facilitate agents training. What we want to achieve with our takeaways is as follows:
>
> *Scenario 1*: If someone is about to collect some datasets for future use, they can, therefore, consider adding some exploration data in the collection pipeline.
>
> *Scenario 2*: If someone has a pre-collected dataset (maybe expert data), and wants to incorporate some online interactions (which might be expensive) for better results, they can use our second remedy to add only minimal interaction data for a good improvement in performance. Doing full online learning is too expensive and takes much longer time in some real-world robotic tasks.
>
> Therefore, we are not misguiding but are providing our understanding of the good data properties in the form of two general strategies. We believe they can help when people meet the corresponding scenarios like those mentioned above.
>
> For **Weakness 3**, that’s true. But these benchmarks are already 4 or 5 years old. Nowadays, people mainly focus on imitation learning with substantial amounts of related papers presented in conferences like CoRL.
>
> For **Weakness 4 & 13 & 14**, we add the two recommended literature in line 36 and line 495. We check the names of authors and remove the editor names. We remove unnecessary literature in the “bulk” references (e.g. in the end of Section 3.3 (line 288), and line 83).
>
> For **Weakness 5**, we omit the symbol of the world model for simplicity because what we want to highlight in Figure 1 is the different variations of data collection among agents. Since we clearly state we are discussing model-based agents everywhere in the paper, it should be clear that the agents have a world model inside.
>
> For **Weakness 6** and **Question 1**, we add pseudocode for Active, Passive, and Tandem agents in Appendix A.1.5 for better clarity. The Active agent is the baseline, implemented by simply running the original DreamerV3 code. The difference in Tandem agent from the Active one is only the model initialization (by a **different seed**, including the policy network). Other factors, like the sampled data batch at each training step, are fixed (copied from Active agents). So it is only removing the self-correction mechanism from the whole training pipeline (no data is generated by its own interaction) and is a more controlled version than Passive agents (since Tandem sees the data batch in the exact same order as Active). Therefore, it can suffer from performance degradation due to the lack of self-correction mechanism.
>
> For **Weakness 7**, yes, we move it to the Appendix, since we only discuss it in the Appendix A.5.
>
> For **Weakness 8**, yes, DreamerV3 collects data online for world model training. However, for policy, it trains only on the imagination rollouts (the starting state is sampled from the replay buffer) by its current world model and policy. Here the dynamics and rewards are estimated by the world model, not given by the true env dynamics. So, if the imagination does not match the real policy rollouts in the environment, it will mislead the policy training (e.g., by a wrong critic) and cause performance degradation.
>
> For **Weakness 9**, we reformulate the hypothesis section, please let us know if you still have confusion about the new version.

---

> > ### Author Response · Authors · 2024-11-23
> >
> > For **Weakness 10**, we believe the performance degradation in Passive/Tandem agents is very counterintuitive. First of all, just like what we discussed in Section 3.5.1 “What is the difference to supervised learning?”, in a typical supervised learning setting (e.g. training a classification network in computer vision), the same dataset should bring similar results, and the model initialization should not affect the model performance. Second, since offline training allows the Passive agent to leverage the good data (i.e., high-reward trajectories, which appear only in the later stage during the Active agent’s training) already from the beginning, it should benefit from this early exposure. However, since, in most cases, neither of these points holds true in model-based RL, it motivates us to investigate this phenomenon further and explore potential explanations.
> >
> > In line 300 (now 311), we wanted to mention a significant dynamics prediction error in the imagination rollouts of the world model due to accumulated errors in multi-step open-loop prediction for all three agents. Previously, we thought this might have contributed to performance degradation. However, with the latest experiments, we found out that degradation is more connected to errors in critics indicated by a clear correlation in Fig. S2. So we delete this paragraph, and replace it with another significant experiment.
> >
> > For **Weakness 12**, the offline dataset is widely available on the Internet, making it inexpensive to access. However, collecting such a dataset independently, especially in real-world robotics tasks, can be costly. Thus, there is no conflict—it's inexpensive in terms of accessibility but expensive from a data collection perspective.
> >
> > Yes, it’s true that allowing interaction will increase performance, but our point is to identify the minimal amount of interaction data required to achieve the same level of improvement in remedy 2.
> >
> > For **other minor points**, we fix them according to the reviewer’s suggestions. The definition of R_t is added to line 137.
> >
> > For **Question 2**, it is “model loss underestimates performance degradation” (as in agents perform better than their model loss suggests).
> >
> > For **Question 3**, it is a different direction from the one discussed in this paper. One can certainly do dataset curation to improve training performance, but in our research, we focus more on the properties of the dataset in a general sense, which is uncurated.

---

> > > ### Comment · Reviewer_tze1 · 2024-11-25
> > >
> > > Dear authors,
> > >
> > > Thank you for your detailed response to what was probably a too long review, which I apologise for. I appreciate you taking the time to go through each of my points in detail.
> > >
> > > > We also update lines 20-23 in the Abstract to better fit our points.
> > >
> > > This is significantly clearer now.
> > >
> > > > For Weakness 2 & 11
> > >
> > > This is a clearer laying out of the goals of the paper, but I believe does not fit with the narrative presented. These concrete aims should, in my opinion, be laid out directly at the start of the paper to motivate the work.
> > >
> > > >  But these benchmarks are already 4 or 5 years old
> > >
> > > Given that there is still significant room for improvement in these benchmarks, as demonstrated by the steady stream of papers being released using D4RL, I do not believe this is a valid reason to dismiss the work.
> > >
> > > > Since we clearly state we are discussing model-based agents everywhere in the paper, it should be clear that the agents have a world model inside.
> > >
> > > The fact that the discussion is related to model based RL is what makes figure 1 so confusing. I will not include this in my consideration of the paper, as I recognise that it may be stylistic/personal, but I believe separating out the agent the world model would make figure 1 significantly clearer.
> > >
> > > > Question 1
> > >
> > > Apologies, I believe I misunderstood the writing. Thank you for clearing that up.
> > >
> > > > For Weakness 9, we reformulate the hypothesis section, please let us know if you still have confusion about the new version.
> > >
> > > This is now much clearer.
> > >
> > > > For Weakness 10, we believe the performance degradation in Passive/Tandem agents is very counterintuitive.
> > >
> > > I have set out why I disagree with this being counterintuitive, and do not find this to be a compelling argument to the contrary. However, I do recognise that there is a difference between finding something 'obvious' and something being *significant*, and so while I still believe a lot of the way the argument is framed is not ideal (i.e., it acts like this is unexpected), I will not hold this against the paper.
> > >
> > > Overall, I believe that the authors have made significant effort to cover many of the relevant issues raised in my review. As such, I have raised my score for contributions from 1 to 2 (since I still believe there is somewhat limited significance) and, correspondingly, increased my review score from 5 to 6. I believe the significance limits this paper from crossing the threshold to a higher score and thus will not increase my score further.

---

### Official Review · Reviewer_EnLo · 2024-11-03

**Soundness:** 2
**Presentation:** 3
**Contribution:** 2
**Rating:** 3
**Confidence:** 3

**Summary:**

This work demonstrates the importance of state novelty in offline model-based reinforcement learning and identifies data composition as a key factor in mitigating performance degradation. The findings on mixed exploratory datasets and self-generated data integration provide valuable insights, suggesting an effective directions for enhancing offline agent robustness and adaptability.

**Strengths:**

1. The paper is overall well written and the main idea is easy to follow.
2. The authors provide sufficient experimental evidence to illustrate the challenges as well as to demonstrate their proposed approach, which enhances both the performance of Passive agents and the efficiency of the online training process.

**Weaknesses:**

1. The novelty of the proposed method is limited. The mixed reward function employed relies on an exploration-exploitation trade-off, which is a well-established framework. For OOD detection, the authors utilize an ensemble approach within the world model, combined with a supervised learning method—both of which are commonly applied in offline RL and lack significant innovation.

2. The approach appears to require task-specific parameter adjustments (e.g., weights for exploration and the OOD ratio), which suggests limited feasibility and generalizability of the method across diverse tasks.

3. The lack of pseudocode for the proposed approach limits clarity. I recommend that the authors include pseudocode in the final version to better illustrate their method and make their contributions more accessible to readers.

**Questions:**

1.  What is the intuition behind constructing a Tandem agent? Training an offline policy that mirrors the action order of the Active agent seems unusual, especially since such an agent is unlikely to be employed in an offline setting prior to online learning.
2. How are the parameters selected for each task? The paper mentions deriving upper bounds based on the performance results of the Active agent. Does this imply that the Active agent must be trained first in order to establish the appropriate parameter ranges? Clarification on this process would enhance understanding.

---

> ### Author Response · Authors · 2024-11-23
>
> We thank the reviewer for providing the valuable comments.
>
> For **Weakness 1 & 2**, please refer to **Q2** in the General Response.
>
> For **Weakness 3**, we have added pseudocodes for the Active, Passive, and Tandem agents, along with the adaptive agents corresponding to the second remedy, in Algorithms 1 and 2 of Appendix A.1.5.
>
> For **Question 1**, the intuition is to investigate the consequence of disabling the self-correction mechanism while keeping many other variables constant (e.g., the data collected and the data batch sampling process at each training step). It is a more controlled setting than Passive agents, making it easier to interpret the results. This experiment allows us to examine whether the difference in model initialization impacts agent training and whether it interacts with the self-correction mechanism. The setting is designed purely for analytical purposes by comparing the Active and Tandem agents. Therefore, it is necessary to run the online learning (Active agent) first to provide the required information for the subsequent Tandem agent's training. We also update Section 2.2 for it to highlight the motivation of Tandem agents compared to other agents.
>
> For **Question 2**, we use a unified threshold for all tasks. The current threshold is chosen by ablation studies in Appendix A.1.6. It is in fact independent of the performance of the Active agents. We look back into the Active agent’s results just to quickly identify a potential search range. It saves us time by providing a good starting point for ablation studies. Accordingly, we update Section A.1.6 to clarify this independence and the purpose of using the upper bound in the ablation studies.

---

### Official Review · Reviewer_g4ug · 2024-11-08

**Soundness:** 3
**Presentation:** 3
**Contribution:** 2
**Rating:** 3
**Confidence:** 4

**Summary:**

This paper examines how different data collection strategies, particularly offline and online data collection, impact the learned world model and policy under the MBRL setting. This investigation is primarily experimental and is conducted using DreamerV3, a state-of-the-art MBRL algorithm. The authors perform extensive experiments across various environments, leading to the following key conclusions:

1.	Insufficient data coverage in MBRL can result in high prediction errors by the model, both in predicting the next state and rewards, on OOD states. Consequently, policies learned from model-generated trajectories may exploit these errors due to the nature of RL.

2.	On-policy samples play a critical role in MBRL. Even with the same sequence of previously collected on-policy training data, minor differences in model initialization can lead to performance degradation. Using purely expert data does not yield good performance either, as distribution shifts occur between the expert policy and the current policy.

3.	Exploration data (or data collected with exploration-enhanced policies) can alleviate issues related to insufficient data coverage. Thus, including exploration rewards during data collection can be beneficial.

Finally, the authors propose two strategies to mitigate performance degradation: incorporating exploration data and adding limited on-policy samples based on the world model loss. While the first approach is straightforward, the second is implemented by monitoring the ratio of world model loss between evaluation and training trajectories. On-policy samples are added once this ratio exceeds a manually set threshold.

Although the paper provides a thorough set of experiments and findings, many of these conclusions have been drawn in prior studies. The primary contribution of this work lies in its experimental validation and in proposing practical remedies for performance degradation.

**Strengths:**

1.	Thorough experimentation: The authors conducted extensive experiments across various environments, providing strong support for the claims made in the paper through the results.

2.	Clear presentation: The paper is well-organized, with definitions and experimental settings clearly presented, making the paper easy to follow.

**Weaknesses:**

1.	Lack of novelty: While the paper offers several analyses, many of these insights exist in previous research. For example, the issue of “lack of self-correction causing OOD errors” is a well-known challenge that has been extensively discussed in both offline RL and offline MBRL algorithms [1][2]. Similarly, the use of on-policy samples or exploration data to achieve broader data coverage has been widely studied in the offline (or offline to online) model-free setting [3][4]. Simply extending these methods to the model-based setting does not add substantial novelty.
2.	Insufficiently effective solution: Despite dedicating a substantial portion of the paper to analyzing issues in offline MBRL methods, the final solution is not effective enough. Whether using exploration data or incorporating on-policy samples, the authors do not provide a principled approach for deciding when and how to incorporate these data. Additionally, the approach relies heavily on hyperparameters in the data strategy, which limits its generalizability across different environments and datasets. \
[1] Off-Policy Deep Reinforcement Learning without Exploration \
[2] MOPO: Model-based Offline Policy Optimization \
[3] Offline-to-Online Reinforcement Learning via Balanced Replay and Pessimistic Q-Ensemble \
[4] Imitation Learning from Imperfection: Theoretical Justifications and Algorithms

**Questions:**

1.	Do the authors have any deeper analysis of the causes and impacts of OOD errors? Currently, most of the analysis either reproduces findings from previous works or directly interprets results without additional insight. A more in-depth analysis beyond compounding error and distribution shift, especially within the MBRL setting, would add value.
2.	Although DreamerV3 is a powerful algorithm, it is unique in its model architecture and algorithmic pipeline compared to other MBRL algorithms. Have the authors conducted any horizontal comparisons across different MBRL algorithms under similar environments or experimental settings?
3.	The authors claim that the world model loss serves as a pessimistic indicator of performance degradation. However, Figure S2 in Appendix A.3.1 shows that while the world model loss sometimes reflects performance degradation, it can be highly unstable due to its reliance on sampled data batches. Additionally, the scale of the world model loss varies significantly across different environments and model architectures. How do the authors interpret the world model loss when it is unstable or has a relatively small scale?

---

> ### Author Response · Authors · 2024-11-23
>
> We thank the reviewer for providing valuable comments.
>
> For **Weakness 1 & 2**, please refer to **Q1 & 2** in the General Response.
>
> For **Question 1**, yes, we conduct two additional experiments, please refer to **Q3** in the General Response.
>
> For **Question 2**, no, we have not. Although DreamerV3 is indeed unique with many tricks inside, the general architecture (Recurrent State-Space Model) and training pipeline (e.g. training policy from imagination) remain common in other state-of-the-art model-based methods. Since the performance degradation issue discussed in our paper primarily arises from the nature of imagination-based offline training, inherent to the training pipeline, this degradation phenomenon should also be evident in other MBRL methods as well. We would like to run TD-MPC2, but it requires a lot of work to separate our current pipeline from DreamerV3. If this would make a big difference in the final evaluation of the paper, we can consider testing it on TD-MPC2 or other algorithms.
>
> For **Question 3**, the instability in world model loss (e.g. fluctuations between large and small values in Tandem agents between timestep 0-200 in Figure S3 of Appendix A.5.1) exactly matches the agent’s oscillation movements. The peak values correspond to the agent’s visiting novel states, which make the agent produce anomaly actions (i.e. oscillation in this case). It goes back to familiar regions but then re-enters novel regions due to its bad policy, creating the oscillation. Since the agent is evaluated on-policy (using its current rollouts), there is no sampling process involved, only how it performs / what it encounters at the moment.
> Generally, from our observation so far, if the (Passive/Tandem) agent is in the familiar region, it should have a low world model loss, having a similar magnitude to the one of Active agent (e.g. in Figure S3). If instability or fluctuation happens, it often means agents encounter novel states.
>
> As for the scales, we introduce the OOD ratio in section 4.2, which normalizes the world model loss by dividing the eval_wm_loss by the train_wm_loss. It makes the metric comparable across different environments and enables the use of a unified threshold. However, in most of our experiments, we only compare the loss within a single environment, rendering this normalized version unnecessary.

---

> > ### Comment · Reviewer_g4ug · 2024-11-26
> >
> > We thank the authors for providing detailed explanations based on the questions.
> >
> > > For Question 2, no, we have not. Although DreamerV3 is indeed unique with many tricks inside, the general architecture (Recurrent State-Space Model) and training pipeline (e.g. training policy from imagination) remain common in other state-of-the-art model-based methods. Since the performance degradation issue discussed in our paper primarily arises from the nature of imagination-based offline training, inherent to the training pipeline, this degradation phenomenon should also be evident in other MBRL methods as well. We would like to run TD-MPC2, but it requires a lot of work to separate our current pipeline from DreamerV3. If this would make a big difference in the final evaluation of the paper, we can consider testing it on TD-MPC2 or other algorithms.
> >
> > While the performance degradation issue primarily arises from the nature of imagination-based offline training, different MBRL algorithms rely differently on imaginations, while some MBRL methods have special designs for alleviating the defect of incorrect imaginations. As this paper is strongly based on experiments, I still recommend the author conduct some experiments on other MBRL methods that different with DreamerV3.
> >
> > > For Question 3, the instability in world model loss (e.g. fluctuations between large and small values in Tandem agents between timestep 0-200 in Figure S3 of Appendix A.5.1) exactly matches the agent’s oscillation movements. The peak values correspond to the agent’s visiting novel states, which make the agent produce anomaly actions (i.e. oscillation in this case). It goes back to familiar regions but then re-enters novel regions due to its bad policy, creating the oscillation. Since the agent is evaluated on-policy (using its current rollouts), there is no sampling process involved, only how it performs / what it encounters at the moment. Generally, from our observation so far, if the (Passive/Tandem) agent is in the familiar region, it should have a low world model loss, having a similar magnitude to the one of Active agent (e.g. in Figure S3). If instability or fluctuation happens, it often means agents encounter novel states.
> >
> > I agree with the explanation of encountering novel states. However, from this perspective, it's hard to say that the loss of world model is a "pessimistic" indicator of performance degradation. The decrease of wm loss doesn't necessarily mean the world model learns better, it could also caused by sampling familiar states. Intuitively, there exists a relation between wm loss and wm's performance, but the intricate relation is hard to describe solely with the value of wm loss on the sampled batches.
> >
> > As for novelty, I acknowledge that the authors investigate performance degradation and its causes in the MBRL setting, which is novel to previous works. However, the major conclusion still lies in the OOD errors, data coverage, and generalization abilities. These problems have been well studied on model-free RL, especially offline model-free RL. Similar analysis of the world model also exists, as is given in the previous review.
> >
> > Finally, thank the authors once again for answering my questions. I’ll keep my scores.

---

### Official Review · Reviewer_3sPu · 2024-11-08

**Soundness:** 3
**Presentation:** 3
**Contribution:** 2
**Rating:** 5
**Confidence:** 3

**Summary:**

This paper presents a large-scale empirical analysis of offline model-based reinforcement learning. The main takeaway is that, encountering OOD states at test time is a main reason for test-time performance degradation of agents. In addition, the authors suggest that this issue can be mitigated by allowing for only interactions.

**Strengths:**

1. The takeaway of this empirical study,  i.e. visiting OOD states causes performance degradation, is clearly presented.
2. The performance of Dreamer-v3 is thoroughly examined in 31 tasks.

**Weaknesses:**

1 The paper confirms a widely studied issue, but it does not reveal novel insights. I agree that, many standard offline RL algorithms (e.g. CQL), approach the issue of visiting OOD states from the perspective of Q values. But I do not think the issue if fundamentally different for offline model-based RL, since in both case, the root cause is that a neural network sees samples not in its training set. See the question 1 & 2 for my suggestion on improving the novelty of this paper.

2. Although aimed to provide insights for practitioners, the three benchmarks used in this paper, DMC, Metaworld, and MinAtar, are in fact not very realistic. In consequence, it is questionable if the same conclusions hold when using more powerful simulators (so it is harder to collect "exploratory" data, since trajectories terminate early).

**Questions:**

1. Since you mention in line 260 that "the paradigm is shifted from ... to the coupling of a world model and a policy network". Could you dissect this coupling? For example, could you provide results of a variant of your Passive agent, which uses the world model of the Active agent but trains its policy from scratch? This is just an example of decoupling the world model and the policy network. Other similar ideas suffice.

2. Could you provide results of Active agent and Passive agent deployed on environments that are slightly different from their training environments? In other worlds, will the superiority of Active agents disappear when their world model cannot accurately predict states?

3. Could you provide results for domains that utilize more realistic simulators, such as Maniskill, LeggedGym, or HumanoidBench?

---

> ### Author Response · Authors · 2024-11-23
>
> We appreciate this author for pointing out the strengths and weaknesses of our paper.
>
> For **Weakness 1**, please refer to  **Q1** in the General Response
>
> For **Question 1**, yes, we carried out the requested experiment. Please refer to **Q3.2** in the General Response.
>
> For **Question 2**, our environment is always randomly initialized, so the current training environment is slightly different from the testing one. Therefore, our results (the superiority of Active agents) already consider this factor. However, if you mean a larger sense of difference (different in task or domain), from our observation, the training of such a policy is difficult and requires a lot of hyperparameter tuning since the default DreamerV3 does not aim for this. Therefore, it is beyond the scope of this project.
>
> For **Question 3** and **Weakness 2**, we acknowledge the potential advantage of testing in a sim (e.g., LeggedGym) for more “realistic” robotic tasks. However, these are not standard RL benchmarks and, therefore, require a substantial amount of effort to integrate our current testing pipeline into them, which is beyond the scope of this project.

---

### Author Response · Authors · 2024-11-23
**General Response**

We thank the reviewers for the detailed comments and your time in reviewing our manuscript. We adjusted our manuscript accordingly in the introduction, hypothesis, and experiments to clearly motivate why our work is needed and our novel insights. Key changes are marked red in pdf.

**Common questions:**

**Q1: General novelty issue: Performance degradation known in other studies and OOD issues not fundamentally different in both model-free and model-based regimes.**

A: We agree with the reviewers that many studies have tried to tackle OOD-induced performance degradation in offline model-based agents. However, our work fundamentally differs by investigating the failure process and explaining the OOD tendency, while other works [1][2] mainly focus on proposing solutions on the premise of such performance degradation. Although there are insightful works [3] in the model-free context (e.g. q-learning), that attribute the degradation process to OOD queries in Q functions during training, there is no evidence that this conclusion can be directly transferred to model-based methods. The coupling of the world model and the policy presents a challenge in interpreting the degradation process. Therefore, the novelty of our paper lies in the effort to explain such a process in model-based RL. Although the way we present the results makes them seem intuitive and “trivial,” the performance degradation in offline MBRL is surprising as it contrasts with the behavior in classical supervised learning (e.g., image classification in CV).

**Q2: Novelty issue in remedies: they are not novel and need further investigation for generalizability**

A: This paper does not aim to propose a novel solution to the degradation issue but rather focuses on understanding the process behind in offline model-based agents. Thereby, we could understand the necessary property of the data that leads to successful offline agent training, which turns out to be sufficient state space coverage. Accordingly, we show that it can be simply achieved by adding a small amount of self-generated data or using exploration data. Although both of the directions have been mentioned in previous studies [3][4], none provide a unified view to highlight the commonality behind them. Besides, they don’t show how much these methods help in practice with various tasks across different domains. Therefore, we believe our conclusion provides novel insights and guidance for future dataset collection. Developing a principled and well-designed approach for these techniques lies beyond the scope of this research, and we leave it to future work.

**Q3: Requesting deeper analysis and insights into the performance degradation**

A: We add two experiments to 1) analyze how the policy is misled during offline training and 2) disentangle the coupling factors of world model and policy.

1) In Appendix A.4, we examine the prediction error of the critics (value functions), which are direct training signals for policy. The results reveal that offline agents exhibit significant prediction errors in critics, highlighting a large discrepancy between the agent’s imagined trajectories and actual rollouts. This discrepancy misguides policy training, leading to visits to OOD states and explaining the OOD tendency observed in offline agents.

2) In Section 3.5.1(details in Appendix A.6), we set the Passive agent’s world model to be a direct copy of the final trained model from the Active agent. For the Tandem agent, at each training step, its world model is updated to be an exact copy of the Active agent's world model at the corresponding training step. In this way, we decouple the influence from a different world model and focus only on a different policy. The results show that both factors affect performance degradation, although their relative importance differs across tasks.

To summarize, the failure process is as follows:
Limited data coverage + different policy —> Misaligned imagination (value functions) —> Wrong update signal to policy —> Agent OOD tendency —> vicious cycle in testing —> bad performance

Accordingly, we reformulate the hypothesis paragraph (Section 3.3) in our paper; please refer to that for a more detailed description of the failure process. We add a brief description of the new experiments into Section 3.4 and 3.5.1.

**Closing**

We appreciate the reviewers' focus on novelty but emphasize that impactful contributions can stem from rigorous analysis of existing algorithms, revealing valuable insights. Our work aims to evaluate these methods in contexts of success and failure, a goal now explicitly clarified in the revised introduction. We kindly request reconsideration of our work in this light.

[1] MOPO: Model-based Offline Policy Optimization

[2] Morel: Model-based offline reinforcement learning

[3] The Difficulty of Passive Learning in Deep Reinforcement Learning

[4] Offline Retraining for Online RL: Decoupled Policy Learning to Mitigate Exploration Bias

---

### Meta-Review · Area_Chair_VGUi · 2024-12-21

**Metareview:**

This paper examines the advantages of online data collection over offline methods in model-based reinforcement learning, finding that online agents outperform offline agents due to challenges with out-of-distribution states. It recommends incorporating additional online interactions and exploration data to enhance the performance of offline agents. Despite of its contributions, this paper still has to be improved on: lack of novel insights into offline model-based reinforcement learning, reliance on unrealistic benchmarks, insufficiently effective solutions, limited novelty in the proposed methods, task-specific parameter adjustments that hinder generalizability, and the absence of pseudocode to enhance clarity.

**Additional Comments On Reviewer Discussion:**

The discussions are somewhat thorough with multi-turns.

---

### Decision · Program_Chairs · 2025-01-22

Reject